# State Size Independent Statistical Error Bound for Discrete Diffusion Models

**Shintaro Wakasugi**[1],   **Taiji Suzuki**[1,2]
[1]The University of Tokyo,   [2]RIKEN AIP
wakasugi-shintaro362@g.ecc.u-tokyo.ac.jp,  taiji@mist.i.u-tokyo.ac.jp

## Abstract

Diffusion models operating in discrete state spaces have emerged as powerful approaches, demonstrating remarkable efficacy across diverse domains, including reasoning tasks and molecular design. Despite their promising applications, the theoretical foundations of these models remain substantially underdeveloped, with the existing literature predominantly focusing on continuous-state diffusion models. A critical gap persists in the theoretical understanding of discrete diffusion modeling: the absence of a rigorous framework for quantifying estimation error with finite data. Consequently, the fundamental question of how precisely one can reconstruct the true underlying distribution from a limited training set remains unresolved. In this work, we analyze the estimation error induced by a score estimation of the discrete diffusion models. One of the main difficulties in the analysis stems from the fact that the cardinality of the state space can be exponentially large with respect to its dimension, which results in an intractable error bound by a naive approach. To overcome this difficulty, we make use of a property that the state space can be smoothly embedded in a continuous Euclidean space that enables us to derive a cardinality independent bound, which is more practical in real applications. In particular, we consider a setting where the state space is structured as a hypercube graph, and another where the induced graph Laplacian can be asymptotically well approximated by the ordinary Laplacian defined on the continuous space, and then derive state space size independent bounds.

## 1   Introduction

Diffusion modeling has demonstrated state-of-the-art performance in learning problems such as creating images (Song et al., 2021; Dhariwal and Nichol, 2021), videos (Ho et al., 2022), and audios (Chen et al., 2020; Kong et al., 2020), drawing significant attention to their applications.

Theoretical studies on diffusion modeling in continuous state spaces have been conducted within the framework of score-based generative modeling (Sohl-Dickstein et al., 2015; Song and Ermon, 2019; Song et al., 2021; Ho et al., 2020; Vahdat et al., 2021). One of the most important characterizations of diffusion modeling is the formulation using stochastic differential equations (SDEs) proposed by Song et al. (2021).

With the advent of SDE formulation, significant efforts have been made to analyze the estimation error between the true distribution and the generated distribution. Lee et al. (2022b) showed that the total variation distance between the two distributions can be bounded by the polynomial order of the score estimation error and the step size with the time discretization of the reverse process. Lee et al. (2022b) assumed the smoothness of the score function and the validity of the log-Sobolev inequality (LSI) for the true distribution, while Chen et al. (2023b) and Lee et al. (2022a) derived error bounds without the LSI condition, Chen et al. (2023a) further relaxed the smoothness assumption. Moreover, Song et al. (2021), Pidstrigach (2022) evaluated error rates under the manifold hypothesis,

which assumes that the true distribution is concentrated on a low-dimensional manifold. De Bortoli et al. (2021) and De Bortoli (2022) derived error bounds under the assumptions of score estimation error bounds at each time step and each point, considering dissipative structures and the manifold hypothesis, respectively.

While the above studies assumed the accuracy for score approximation, Oko et al. (2023) developed a theoretical framework that derives the approximation error of the score function when the true density belongs to a Besov space. Their work combined function approximation theory from deep learning with the approximation theory of diffusion modeling and employed concentration inequalities to establish score estimation error bounds. Building upon this seminal work, several refined theoretical analyses have been proposed, including relaxation of the lower bound condition for the density (Zhang et al., 2024), analyses under the manifold assumption on the support of the data distribution (Azangulov et al., 2024), statistical guarantees for reflected diffusion models (Holk et al., 2024), and minimax optimality of the probability flow ODE (Cai and Li, 2025).

Recently, diffusion modeling for discrete states has also gained attention (Hoogeboom et al., 2021; Austin et al., 2021; Richemond et al., 2022; Meng et al., 2022; Sun et al., 2023; Santos et al., 2023; Lou et al., 2024). Notable advancements have been made in learning problems with discrete structures, such as natural language processing (Austin et al., 2021; He et al., 2023; Wu et al., 2023), molecular design (Zhang et al., 2023; Gruver et al., 2023; Campbell et al., 2024; Lee et al., 2025), graph generation (Niu et al., 2020; Shi et al., 2020; Vignac et al., 2023), and segmentation (Zbinden et al., 2023). In addition, in areas where continuous diffusion modeling performs well, such as image (Hu et al., 2022; Zhu et al., 2023) and audio generation (Yang et al., 2023), discrete diffusion models have been shown to efficiently infer multimodal generation problems conditioned on discrete structures like text.

As for the theoretical analysis of discrete diffusion modeling, Campbell et al. (2022) analyzed total variation distance based on Markov chains, and Chen and Ying (2024) reduced the error under the condition that the discrete state space is restricted to the vertices of a hypercube. Ren et al. (2025) proposed a more general approximation theory for discrete diffusion characterized by a Poisson random measure with evolving intensity, allowing discrete diffusion modeling to be formulated as stochastic integrals similar to the theory of continuous diffusion modeling.

However, the theoretical analysis of score estimation error in discrete diffusion modeling remains unexplored. In this study, we derive score estimation error bounds for discrete diffusion models by applying the function approximation theory of neural networks and concentration inequalities, which were previously used in the score estimation theory of continuous diffusion modeling (Oko et al., 2023). Our main contributions are summarized as follows:

- First, we develop a theoretical framework for bounding the score estimation error in discrete diffusion models. Unlike continuous diffusion models that rely on $L^2$ loss in score matching, our analysis handles the Bregman divergence loss that naturally arises in discrete diffusion, and we rigorously control the estimation error using the Hellinger distance.

- Second, we introduce a novel approach to achieve state-size independent error bounds by embedding the discrete space $\mathbb{X}$ into $\mathbb{R}^d$ and approximating the eigenvectors of the graph Laplacian by functions in an anisotropic Besov space. This enables the use of advanced function approximation results for deep ReLU networks. Under mild regularity assumptions, we show that the error bound depends only polylogarithmically on the number of discrete states $M$, which nearly achieves the optimal rate conjectured in Ren et al. (2025).

- Third, we demonstrate that this framework can be instantiated in concrete settings such as the hypercube $[0,1]^D$ and graph-based diffusion processes on smooth manifolds. In both examples, we show that the eigenvectors of the transition matrix admit efficient approximations.

## 2   Preliminary

Here, we introduce discrete diffusion models and prepare some technical matters for our theoretical analysis. Before introducing the discrete diffusion models, we briefly review the continuous state diffusion models. The continuous diffusion models consist of two stochastic processes, the forward process and the reverse process, so that the model can generate data whose distribution is sufficiently close to the true distribution $p_0$ on $\mathbb{R}^d$. The forward process $\{X_t\}_{t \geq 0}$ on $\mathbb{R}^d$ is formulated as the

following Ornstein-Uhlenbeck (OU) process:

$$X_0 \sim p_0, \quad \mathrm{d}X_t = -X_t \mathrm{d}t + \sqrt{2}\mathrm{d}B_t,$$

where $\{B_t\}_{t \geq 0}$ represents a $d$-dimensional standard Brownian motion. Under certain assumptions on the initial distribution $p_0$ (Haussmann and Pardoux, 1986; Cattiaux et al., 2023), the distribution of $X_t$, denoted by $p_t$, converges exponentially to the standard normal distribution as $t \to \infty$. The reverse process $\{Y_t\}_{0 \leq t \leq T}$ ($T \geq 0$) defined by the following stochastic process *trace-backs* the forward process:

$$Y_0 \sim p_T, \ \mathrm{d}Y_t = (Y_t + 2\nabla \log p_{T-t}(Y_t))\mathrm{d}t + \sqrt{2}\mathrm{d}B_t.$$

That is, the distribution of $Y_{T-t}$ coincides with $p_t$ for $0 \leq t \leq T$, and thus we can generate samples from the target distribution $p_0$ by sampling $Y_T$ via the reverse process. However, we do not know the initial distribution of the reverse process $p_T$ and the *score function* $\nabla \log p_t$ because both of them are dependent on the unknown true distribution $p_0$. Here, the initial distribution $p_T$ can be replaced by the standard normal distribution for sufficiently large $T$ since $p_t$ converges to the standard normal exponentially fast, and the score function $\nabla \log p_t(Y_t)$ can be estimated by least squares score matching using finite-size training data. These approximations, along with the time discretization of the process, induce errors in the resulting distribution generated by the model. Chen et al. (2023a) obtained an upper bound on this error in terms of Kullback–Leibler divergence (KL-divergence). Their result implies that the main part of the error is the estimation error of the score function $\nabla \log p_t$. Based on this observation, Oko et al. (2023) derived the following bound on the score estimation error.

**Proposition 1** (Oko et al. (2023), informal). *Under some smoothness assumptions on $p_0$, a score matching estimator $\widehat{s}_t$ obtained on a deep neural network model with an appropriate network size, can achieve*

$$\mathbb{E}_{x_{0,i}} \left[ \int_\delta^T \mathbb{E}_{x_t \sim p_t} \left[ \|\nabla \log p_t(x_t) - \widehat{s}_t(x_t)\|^2 \right] \mathrm{d}t \right] \lesssim n^{-\frac{2s}{d+2s}} \log^{18} n. \tag{1}$$

*Here, $s$ represents the smoothness parameter of the Besov space to which $p_0$ belongs, and $\delta > 0$ is a sufficiently small end-point time.*

This theorem implies that diffusion models with an appropriately designed score-function estimator can achieve the minimax optimal rate $n^{-\frac{2s}{d+2s}}$ to estimate the target distribution with smoothness $s$. In this work, we establish an analogue of this result for discrete diffusion models.

## 2.1 Discrete diffusion modeling

Discrete diffusion modeling is defined over a finite set $\mathbb{X}$ instead of $\mathbb{R}^d$. Let $M := |\mathbb{X}|$, and consider estimating the probability mass vector $p_0 \in \Delta^M$, where $\Delta^M := \{p \mid \sum_{x \in \mathbb{X}} p(x) = 1, \ p \in \mathbb{R}^M_{\geq 0}\}$. We assume that each $x \in \mathbb{X}$ has a vector representation $\iota(x) \in \mathbb{R}^D$. We identify this vector representation with $x$ and use the same notation $x$ for both meanings. A typical situation is $\mathbb{X} = \{0,1\}^D$ where $\iota(x) = (x_1, \ldots, x_D) \in \mathbb{R}^D$, and another example is one-hot-vector representation $\mathbb{X} = \{x \in \{0,1\}^M \mid \sum_{i=1}^M x_i = 1\}$.

**Forward process:** In the forward process of discrete diffusion modeling, the distribution $\{p_t\}_{t \geq 0}$ at each time step follows the master equation of the following Markov process:

$$\frac{\mathrm{d}p_t}{\mathrm{d}t} = Q_t p_t, \tag{2}$$

where $Q_t \in \mathbb{R}^{M \times M}$ is the transition rate matrix satisfying $Q_t(x,x) = -\sum_{y \neq x} Q_t(y,x)$ ($\forall x \in \mathbb{X}$) and $Q_t(x,y) \geq 0$ ($\forall x \neq y \in \mathbb{X}$). If $\pi$ denotes the stationary distribution of the Markov process (2), the following equation analogous to the continuous diffusion model holds (Bobkov and Tetali, 2006):

$$\mathrm{KL}(p_t||\pi) \leq \exp(-\rho(Q)t)\mathrm{KL}(p_0||\pi). \tag{3}$$

Here, $\rho(Q)$ is the modified log-Sobolev constant(Bobkov and Tetali, 2006) defined as

$$\rho(Q) := \inf \left\{ \frac{\mathcal{E}_\pi(f, \log f)}{\mathrm{Ent}_\pi(f)} \mid f : \mathbb{X} \mapsto \mathbb{R}, \ \mathrm{Ent}_\pi(f) > 0 \right\},$$

where $\mathrm{Ent}_\pi(f) := \mathbb{E}_\pi[f \log f] - \mathbb{E}_\pi[f] \log \mathbb{E}_\pi[f]$ and $\mathcal{E}_\pi(f,g) = \mathbb{E}_\pi[f Q^\top g]$.

**Reverse process:** The reverse process $\{q_t\}_{0 \le t \le T} = \{p_{T-t}\}_{0 \le t \le T}$ can be formulated by using another transition rate matrix $\overline{Q}_t$ as follows (Kelly, 2011):

$$q_0 = p_T, \quad \frac{\mathrm{d}q_t}{\mathrm{d}t} = \overline{Q}_{T-t}q_t \ \ (0 \le t \le T), \tag{4}$$

where $\overline{Q}_t(y, x) = \frac{p_t(y)}{p_t(x)}Q_t(x, y)$ and $\overline{Q}_t(x, x) = -\sum_{y \ne x} \overline{Q}_t(y, x)$. Then, it is known that $q_t = p_{T-t}$ ($0 \le t \le T$). Although this reverse process only gives the ODE of the probability mass function $q_t$, its particle implementation can be given by the $\tau$-*leaping algorithm* (Campbell et al., 2022; Ren et al., 2025). This algorithm expresses the reverse process as a stochastic integral and applies the Euler-Maruyama scheme, similar to the continuous case. The transformation into a stochastic integral is performed using a Poisson random measure with evolving intensity (Protter, 1983). For simplicity, we consider the case where $Q_t$ is time-homogeneous ($Q_t = Q$ holds for all $t \ge 0$).

**Proposition 2** (Ren et al. (2025))**.** *The reverse process* (4) *can be expressed as the following stochastic integral defined by a Poisson process $N[\mu]$ with respect to an intensity function $\mu$ (see Definition 2 in Appendix B for its definition):*

$$y_t = y_0 + \int_0^t (y - y_{t-})N[\mu](\mathrm{d}t, \mathrm{d}y), \quad \mu_t(y) = s_{T-t}^{\circ}(y_{t-}, y)\widetilde{Q}(y_{t-}, y).$$

*where $X_{t-}$ denotes the left limit of $X_t$ and $\widetilde{Q}$ denotes the matrix $Q$ with the diagonal elements set to 0.*

**Score estimation:** However, to implement the $\tau$-leaping algorithm, we need to know the score function $s_t^{\circ}(x, y) := \frac{p_t(y)}{p_t(x)}$. We approximate it using a score network $s : \mathbb{X}^2 \times \mathbb{R} \to \mathbb{R}$ $((x, y, t) \mapsto s_t(x, y))$ in a deep neural network model $\mathcal{F}$. The score network $s$ can be estimated via score matching analogous to the continuous state diffusion models, but we need to account for the non-negativity constraint of the score function. For that purpose, the denoising score entropy is employed instead of $L^2$ loss (Lou et al., 2024):

$$s \in \arg\min_{s \in \mathcal{F}} \int_0^T \mathbb{E}_{x \sim p_t} \left[ \sum_{y \ne x} \mathrm{BR}_K(s_t(x, y) \| s_t^{\circ}(x, y)) \cdot s_t^{\circ}(x, y)Q(x, y) \right] \mathrm{d}t$$

$$= \arg\min_{s \in \mathcal{F}} \int_0^T \mathbb{E}_{x_0 \sim p_0} \left[ \mathbb{E}_{x \sim p_t(\cdot|x_0)} \left[ \sum_{y \ne x} \mathrm{BR}_K(s_t(x, y) \| s_t^{\circ}(x, y \mid x_0)) \cdot s_t^{\circ}(x, y \mid x_0)Q(x, y) \right] \right] \mathrm{d}t.$$

Here $s_t^{\circ}(x, y \mid x_0) := \frac{p_t(y|x_0)}{p_t(x|x_0)}$ and BR denotes the Bregman divergence defined by

$$\mathrm{BR}_f(s^* \| s) := f(s^*) - f(s) - \partial f(s)(s^* - s),$$

for a strictly convex function $f$, where we employ a particular choice $K(x) := x - \log x$ for the convex function $f$. In training the neural network, analogous to continuous diffusion models, we approximate the expectation over $x_0 \sim p_0$ by empirical distribution defined by the training data $D_n := \{x_i\}_{i=1}^n$ ($x_i \overset{\text{i.i.d.}}{\sim} p_0$) with size $n$, and we seek $\widehat{s} \in \mathcal{F}$ that minimizes this empirical loss. We define the loss function $\ell$ as

$$\ell_s(x_0) := \int_\delta^T \mathbb{E}_{x_t \sim p_t(\cdot|x_0)} \left[ \sum_{y \ne x_t} \mathrm{BR}_K(s_t(x_t, y) \| s_t^{\circ}(x_t, y \mid x_0)) \cdot s_t^{\circ}(x_t, y \mid x_0)Q(x_t, y) \right] \mathrm{d}t.$$

Then, the empirical loss $\widehat{L}(s)$ can be written as $\widehat{L}(s) := \frac{1}{n}\sum_{i=1}^n \ell_s(x_i)$. As we have stated above, we find the empirical risk minimizer $\widehat{s}$ in the set of deep neural networks that is defined as

$$\Phi(L, W, S, B) := \{(A^{(L)}\eta(\cdot) + b^{(L)}) \circ \cdots \circ (A^{(1)}x + b^{(1)}) \mid$$

$$A^{(i)} \in \mathbb{R}^{W_{i+1} \times W_i}, b^{(i)} \in \mathbb{R}^{W_{i+1}}, \ \textstyle\sum_{i=1}^L (\|A^{(i)}\|_0 + \|b^{(i)}\|_0) \le S, \max_i \|A^{(i)}\|_\infty \vee \|b^{(i)}\|_\infty \le B\},$$

where $L$ represents the depth, $W = (W_i)_{i=1}^L$ represents the width with $W_{L+1} = 1$, $B$ represents the sparsity, and $B$ is a bound on the norm of parameters. Here, the activation function $\eta(\cdot)$ is given by $\eta(x) = \mathrm{ReLU}(x) := \max(x, 0)$. The function class to which the score network belongs is defined

---
**Algorithm 1** Implementation of discrete diffusion modeling by $\tau$-leaping
---
1: **Input:** $\widehat{y}_0 \sim \pi$, time discretization $\{t_k\}_{k \in [0,K]}$ ($t_0 = 0, t_K = T - \delta$), intensity function $\hat{\mu}_t$, score network $\widehat{s}_t$
2: **Output:** sample from $\widehat{y}_{t_K} \sim \widehat{q}_{T-\delta}$
3: **for** $n = 0$ to $K - 1$ **do**
4:     $\widehat{y}_{t_{n+1}} \leftarrow \sum_{y \in X}(y - \hat{y}_{t_n})\mathcal{P}(\widehat{\mu}_{t_n}(y)(t_{n+1} - t_n));$
5: **end for**
---

as $\mathcal{F} := \{s \in \Phi(L, W, S, B) \mid s_t(x, y) \in [1/R, R] \ (\forall t, x, y)\}$ with a hyper-parameter $R \geq 1$ (see Assumption 3)[1].

We denote the expectation of a measurable function $f$ with respect to $x_0 \sim p_0$ as $Pf$, and its empirical distribution as $P_n f$, i.e., $Pf := \mathbb{E}_{x_0 \sim p_0}[f]$ and $P_n f := \frac{1}{n}\sum_{i=1}^{n} f(x_i)$. Accordingly, the expected loss $L(s)$ and empirical loss $\widehat{L}(s)$ can be expressed as

$$L(s) := \mathbb{E}_{x_0 \sim p_0}[\ell_s(x_0)] = P(\ell_s), \quad \widehat{L}(s) := \frac{1}{n}\sum_{i=1}^{n}\ell_s(x_i) = P_n(\ell_s),$$

Then, the empirical risk minimizer on the deep neural network model $\mathcal{F}$ is given by

$$\widehat{s} \in \arg\min_{s \in \mathcal{F}} P_n(\ell_s).$$

Once we have obtained an estimator $\widehat{s}$, we can define the corresponding intensity function $\widehat{\mu}_t$ with time discretization $(t_k)_{k=0}^{K}$, with $t_0 = 0$ and $t_K = T - \delta$, as

$$\widehat{\mu}_{\lfloor t \rfloor}(y) = \widehat{s}_{T-\lfloor t \rfloor}(\widehat{y}_{\lfloor t \rfloor -}, y)\widetilde{Q}(\widehat{y}_{\lfloor t \rfloor -}, y),$$

where $\lfloor t \rfloor = t_k$ for $t \in [t_k, t_{k+1})$. Moreover, due to Eq. (3), the initial distribution of the reverse process $q_0$ can be replaced by $\pi(\simeq p_T)$. Then, the $\tau$-leaping algorithm with our estimate $\widehat{s}$ can be implemented as in Algorithm 1.

## 2.2 Technical assumptions and theoretical tools for the error analysis

To derive the discrepancy between the generated distribution and the true target distribution for the estimated discrete diffusion model implemented by $\tau$-leaping algorithm (Alg. 1), we prepare some technical tools.

**Assumption 1.** *The transition rate matrix $Q$ is symmetric, and there exist positive constants $C, \underline{D}$, and $\overline{D}$ such that $Q(x, y) < C$, $\underline{D} < -Q(x, x) < \overline{D}$.*

**Assumption 2.** *There exists a lower bound $\rho > 0$ for the modified log-Sobolev constant $\rho(Q)$.*

**Assumption 3.** *There exists $R \geq 1$ such that the true score function and our model satisfy $s_t^\circ(x, y) \in [1/R, R]$ and $\widehat{s}_t(x, y) \in [1/R, R]$.*

**Assumption 4.** *There exists $\gamma \in [0, 1]$ such that for any $y \in \mathbb{X}$ satisfying $Q(x_{t-}, y) > 0$, the following holds: $\left| \frac{p_t(x_{t-})Q(x_t, y)}{p_t(x_t)Q(x_{t-}, y)} - 1 \right| \lesssim 1 \vee t^{-\gamma}$.*

Assumption 1 is a natural condition for discrete diffusion models which ensures the regularity of the rate matrix (Ren et al., 2025). Assumption 2 guarantees the exponential convergence of the forward process in discrete diffusion models, serving a role analogous to functional inequalities in the continuous case (Bakry et al., 2014). Assumption 3 imposes boundedness on the score function, which mirrors common assumptions made in continuous-state diffusion processes (Chen et al., 2023a). Assumption 4 corresponds to the Lipschitz continuity of the score function in continuous diffusion models (Chen et al., 2023b,a), ensuring sufficient regularity for the analysis. Under these assumptions, the following result holds.

---

[1]The function value restriction can be practically implemented as $\Phi' = \max\{\min\{\Phi, R\}, R^{-1}\}$ where $\min$ and $\max$ can be realized by ReLU activation.

**Proposition 3** (Ren et al. (2025)). [2] *In the $\tau$-leaping algorithm, suppose the time discretization $\{t_k\}_{k\in[0,K]}$ satisfies $t_{k+1} - t_k \leq \kappa(1 \vee (T - t_{k+1})^{1+\gamma-\eta})$ for some $\eta > 0$ and assume that*

$$\sum_{k=0}^{K-1}(t_{k+1}-t_k)\mathbb{E}\Big[\sum_{y\neq x_{t_k-}}\mathrm{BR}_K(\widehat{s}_{T-t_k}(x_{t_k-},y)\|s^\circ_{T-t_k}(x_{t_k-},y))s^\circ_{T-t_k}(x_{t_k-},y)Q(x_{t_k-},y)\Big]\leq\varepsilon_{\mathrm{sc}},$$

*for the score network $\widehat{s}_t(x,y)$. Here, we suppose that $\gamma < \eta \lesssim 1 - T^{-1}$ for $\gamma < 1$, and $\eta = 1$ for $\gamma = 1$, and it holds that*

$$T = \mathcal{O}\left(\frac{\log(\varepsilon_{\mathrm{sc}}^{-1}\log|\mathbb{X}|)}{\rho}\right), \ \kappa = \mathcal{O}\left(\frac{\varepsilon_{\mathrm{sc}}\rho}{\bar{D}^2\log(\varepsilon_{\mathrm{sc}}^{-1}\log|\mathbb{X}|)}\right), \ \delta = \begin{cases} 0 & (\gamma < 1), \\ \Omega\left(e^{-\sqrt{T}}\right) & (\gamma = 1). \end{cases}$$

*Then, under Assumption 1 to 4, the following error bound holds*

$$\mathrm{KL}(p_\delta\|\widehat{q}_{T-\delta}) \lesssim \underbrace{\exp(-\rho T)\log M}_{(i)} + \underbrace{\bar{D}^2\kappa T}_{(ii)} + \underbrace{\varepsilon_{\mathrm{sc}}}_{(iii)}.$$

Similar to continuous diffusion modeling, the upper bound on the error consists of three terms. (i) corresponds to the error from the convergence rate of the forward process, as shown in Eq. (3). (ii) arises from time discretization. (iii) represents the estimation error of the score network. Although the Girsanov's theorem cannot be applied to discrete diffusion models, another similar proposition can be derived from the characterization of a Poisson random measure with evolving intensity (Ren et al., 2025), which allows for the evaluation of the estimation error. However, unlike continuous diffusion models, the sample size or neural network parameter size required to satisfy the estimation error assumption has not been examined. Hence, the aim of this paper is to show an upper bound of $\varepsilon_{\mathrm{sc}}$ analogous to Eq. (1).

## 3 Naive estimation error bound of discrete diffusion models

Here, we derive an estimation error bound of the score estimation network in a rather naive way. In our analysis, the key step is to derive an upper bound on the *Hellinger distance* between the empirical risk minimizer $\widehat{s}$ and the true score function $s^\circ$, defined by

$$h(\widehat{s}_t, s^\circ_t) := \left(\mathbb{E}_{x_t\sim p_t}\left[\sum_{y\neq x_t}\left(\sqrt{\widehat{s}_t(x_t,y)} - \sqrt{s^\circ_t(x_t,y)}\right)^2 Q(x_t,y)\right]\right)^{1/2}.$$

First, we give a bound on this Hellinger distance under a naive setting.

**Theorem 1.** *Under Assumption 1 to 4, and the same parameter settings as Proposition 3, if the network size is set as $L = \mathcal{O}(\log^2(mM))$, $W = \tilde{\mathcal{O}}(M)$, $S = \tilde{\mathcal{O}}(M)$, $B = \tilde{\mathcal{O}}(M^2)$ and $T = \mathcal{O}(\log(M \vee n))$, then with probability at least $1 - 2e^{-t}$ for $t \geq 1$, it holds that*

$$\int_\delta^T h^2(\widehat{s}_t, s^\circ_t)\mathrm{d}t = \mathcal{O}\left(\frac{Mt}{n}\log^8(M \vee n)\right).$$

The formal proof of Theorem 1 is provided in Appendix C . By Proposition 3 and Theorem 1, we obtain the following result showing an estimation error bound for the estimated distribution.

**Theorem 2.** *Suppose that the same condition as that of Theorem 1 holds. Then, the following estimation error bound holds with probability $1 - 2e^{-t}$ for $t \geq 1$:*

$$\mathrm{KL}(p_\delta\|\widehat{q}_{T-\delta}) \lesssim \frac{Mt}{n}\log^8(M \vee n).$$

One of the biggest difficulties in deriving this bound is the requirement to carefully treat the Bregman divergence in contrast to the continuous diffusion models where we could directly deal with the $L^2$-norm. The difficulty can be overcome by noticing that the Bregman divergence can be lower bounded by the Hellinger distance. By combining this notion with the so-called *peeling device*

---

[2]It was later pointed out that the $\tau$-leaping update can become ill-defined for non-ordinal data when multiple-jumps appear in a single time-window, which we were not aware of at the time of writing. Our theoretical results do not rely on this aspect and therefore remain valid.

van de Geer (2000), we achieve $\mathcal{O}(1/n)$ rate of convergence with respect to the sample size $n$, which significantly improves upon the standard Rademacher complexity bound that only provides an $\mathcal{O}(1/\sqrt{n})$ bound. An additional technical contribution is to show how ReLU neural networks can effectively approximate the score function $s_t$.

This bound holds under minimal assumptions. On the other hand, if the state space is a product space such as $\mathbb{X} = \{0, 1\}^D$ (which often happens in real applications), then $M$ can be *exponentially large* with respect to the dimension $D$. To overcome this difficulty, we consider a situation where the discrete state space $\mathbb{X}$ can be embedded into a Euclidean space $\mathbb{R}^d$ and the eigenvectors corresponding to the Markov transition operator $Q$ can be well approximated by a smooth function on $\mathbb{R}^d$. By doing so, we obtain a significantly improved bound as seen in the next section.

## 4 State size independent error analysis with continuous space embedding

As we mentioned above, we aim to improve the estimation error by explicitly considering an embedding from the discrete space $\mathbb{X}$ into a Euclidean space $\mathbb{R}^D$. Through the embedding, we may approximate the eigenvectors of $Q$ by functions defined on $\mathbb{R}^D$ enabling the application of function approximation theory for smooth functions defined on Euclidean spaces. Especially, the *anisotropic Besov space* is a useful function class that covers a wide range of functions with smoothness.

**Anisotropic Besov space.** Here, we begin by defining anisotropic Besov spaces. Let $\Omega = [0, 1]^d$. For a function $f : \Omega \to \mathbb{R}$, we define its $L^p$-norm as

$$\|f\|_p := \|f\|_{L^p(\Omega)} := \begin{cases} (\int_\Omega |f|^p dx)^{1/p} & (0 < p < \infty), \\ \sup_{x \in \Omega} |f(x)| & (p = \infty). \end{cases}$$

For $\beta \in \mathbb{R}^d_{++}$, let $|\beta| := \sum_{j=1}^d |\beta_j|$. The $r$-th order finite difference in the direction $h \in \mathbb{R}^d$ is defined as: $\Delta_h^r(f)(x) := \Delta_h^{r-1}(f)(x + h) - \Delta_h^{r-1}(f)(x)$ and $\Delta_h^r(f)(x) := f(x)$, for $x + rh \in \Omega$; otherwise, we define $\Delta_h^r(f)(x) = 0$.

**Definition 1** (Anisotropic Besov Space). *Let $0 < p, q \leq \infty$, $\beta = (\beta_1, \ldots, \beta_d)^\top \in \mathbb{R}^d_{++}$, $r := \max_i \lfloor \beta_i \rfloor + 1$. Then the Besov semi-norm is defined as*

$$|f|_{B^\alpha_{p,q}} := \begin{cases} \left( \sum_{k=0}^\infty [2^k w_{r,p}(f, (2^{-k/\beta_1}, \ldots, 2^{-k/\beta_d}))]^q \right)^{1/q} & (q < \infty), \\ \sup_{k \geq 0} 2^k w_{r,p}(f, (2^{-k/\beta_1}, \ldots, 2^{-k/\beta_d})) & (q = \infty), \end{cases}$$

*where $w_{r,p}$ is the $r$-th order modulus of smoothness defined by $w_{r,p}(f, t) := \sup_{h \in \mathbb{R}^d : |h_i| \leq t_i} \|\Delta_h^r(f)\|_p$. The anisotropic Besov space $B^\beta_{p,q}(\Omega)$ is defined as $B^\beta_{p,q}(\Omega) := \{ f \in L^p(\Omega) \mid \|f\|_{B^\beta_{p,q}} := \|f\|_p + |f|_{B^\beta_{p,q}} < \infty \}$. The unit ball of the anisotropic Besov space is denoted by $U(B^\beta_{p,q})$.*

The harmonic mean of the components of $\beta$, which is given by $\widetilde{\beta} := (\sum_{j=1}^d 1/\beta_j)^{-1}$, plays an important role in evaluating the approximation error by ReLU deep neural networks. It is known that the Hölder class and the Sobolev class with $p = 2$ are special cases of anisotropic Besov spaces Triebel (1983, 2011). In the following, we consider a situation where the eigenvectors of $Q$ can be well approximated by a function in a Besov space on the continuous space.

### 4.1 Assumpitons

We now summarize the additional assumptions required for the analysis.

**Assumption 5.** *Let $0 < \varepsilon < 1$. Suppose the orthonormal eigenvectors $U = (u_1, \ldots, u_M)$ of the graph Laplacian $L = -Q$ satisfy $u_j(x) = \mathcal{O}(1/\sqrt{M})$ for all $x \in \mathbb{X}$ and the initial distribution $p_0$ can be expanded as:*

$$p_0(x) = \sum_{j=1}^M c_j u_j(x) \quad (\forall x \in \mathbb{X}).$$

*Assume that for each $j = 1, \ldots, M$, there exists a function $\sqrt{M} u_j^* : \mathbb{X} \to [0, 1]$ satisfying $|u_j^*(x) - u_j(x)| \leq \varepsilon/\sqrt{M}$ and representable as $u_j^*(x) = h_j(Px)$, where $h_j \in \gamma_j U(B^\beta_{p,q})$ with $\gamma_j > 0$ and $P \in \mathbb{R}^{d \times D}$ are projection matrices for all $j$. Moreover, assume $\|P\|_\infty = \mathcal{O}(1)$ and $\widetilde{\beta} > 1/p$.*

**Assumption 6.** *For each $j = 1, \ldots, M$, the expansion coefficient satisfies $|c_j| \lesssim |c_1| \cdot j^{-s}$ and the Besov norms of $h_j$ satisfy $|\gamma_j| \lesssim j^{\gamma}$ where $s > 0$ and $\gamma \geq 0$.*

Assumption 5 is a technical condition that enables the application of function approximation theories in the anisotropic Besov space (Suzuki and Nitanda, 2021) (see also Suzuki (2019)). The factor $\mathcal{O}(1/\sqrt{M})$ in $u_j$ stems from the normalization of orthonormal eigenvectors, i.e., $\sum_{x \in \mathbb{X}} u_j(x)^2 = 1$. Assumption 6 imposes a polynomial decay condition on $c_j$, which is a standard regularity assumption in nonparametric statistics, particularly in the analysis of kernel methods (Caponnetto and De Vito, 2007; Ying and Pontil, 2008; Dieuleveut and Bach, 2016). The condition on $\gamma_j$ reflects the increasing complexity of the basis functions $h_j$. Similar assumptions are common in the analysis of eigenfunctions of the Laplacian operator in continuous settings.

### 4.2 State space size independent error bound

**Theorem 3.** *Assume that Assumption 5 and 6 as well as Assumption 1 to 4 hold, and if the network size is set as $L = \mathcal{O}(\log^2(M))$, $W = \tilde{\mathcal{O}}(M)$, $S = \tilde{O}(M)$, $B = \tilde{O}(M^2)$ and $T = \mathcal{O}(\log(M \vee n))$, the following estimation error bound holds with probability $1 - 2e^{-t}$:*

$$
\int_{\delta}^{T} h^2(\hat{s}_t, s_t^{\circ}) \mathrm{d}t = \begin{cases} \mathcal{O}\left( \left( \lambda_2^{-\frac{2\tilde{\beta}}{(1+2\tilde{\beta})(s-1)+\tilde{\beta}}} n^{-\frac{2\tilde{\beta}(s-1)}{(1+2\tilde{\beta})(s-1)+\tilde{\beta}}} + \left( \frac{\varepsilon}{\lambda_2} \right)^2 + \frac{t}{n} \right) \log^8(M \vee n) \right) & (s - \gamma \geq 1), \\[2mm] \mathcal{O}\left( \left( M^{\frac{2(1-(s-\gamma)+\tilde{\beta})}{1+2\tilde{\beta}}} n^{-\frac{2\tilde{\beta}}{1+2\tilde{\beta}}} + M^2 \left( \frac{\varepsilon}{\lambda_2} \right)^2 + \frac{t}{n} \right) \log^8(M \vee n) \right) & (s < 1), \\[2mm] \mathcal{O}\left( \left( \lambda_2^{-\frac{2(1-(s-\gamma)+\tilde{\beta})}{(1+2\tilde{\beta})(s-1)+(1-(s-\gamma)+\tilde{\beta})}} n^{-\frac{2\tilde{\beta}(s-1)}{(1+2\tilde{\beta})(s-1)+(1-(s-\gamma)+\tilde{\beta})}} + \frac{t}{n} \right. \right. \\ \left. \left. + \varepsilon^2 \left( \frac{n^{\tilde{\beta}(1-(s-\gamma))}}{\lambda_2^{2\gamma(1+2\tilde{\beta})+2(1-(s-\gamma)+\tilde{\beta})}} \right)^{\frac{1}{(1+2\tilde{\beta})(s-1)+(1-(s-\gamma)+\tilde{\beta})}} \right) \log^8(M \vee n) \right) & (otherwise). \end{cases}
$$

The proof of Theorem 3 is given in Appendix D. The estimation error bound in Theorem 3 shows that for $s \geq 1$, the dependence on $M$ is only polylogarithmic, successfully removing any polynomial dependence on $M$. This aligns with the optimal rate conjectured in Ren et al. (2025) for discrete diffusion models. Even when $\gamma + 1/2 < s < 1$, the exponent on $M$ remains below 1, yielding an improved convergence rate over Theorem 1.

As for the dependence on the sample size $n$, the bound does not explicitly depend on the embedded dimension $d$, which is desirable. Moreover, when $s < 1$, the rate recovers the optimal rate $n^{-2\tilde{\beta}/(1+2\tilde{\beta})}$ derived by Suzuki and Nitanda (2021), but can be dependent on $M$ polynomially. This is because estimation errors for $\mathcal{O}(M)$-basis functions affect the final results when the decrease of coefficients is slow. On the other hand, when $s$ is large, we may "cut-off" redundant basis functions so that we mitigate the dependency on $M$ to poly-log order while the rate with respect to $n$ becomes a bit slower instead.

Finally, combining Proposition 3 and theorem 3 yields an upper bound on the distribution estimation error. Here, we let the right hand side of the bound in Theorem 3 be denoted by $\Xi_{n,t}$.

**Theorem 4.** *Suppose that the same condition as that of Theorem 3 holds. Then, the following error bound holds with probability $1 - 2e^{-t}$:*

$$
\mathrm{KL}(p_\delta \| \hat{q}_{T-\delta}) \lesssim \Xi_{n,t}.
$$

*Example 1: Hypercube $\{0, 1\}^D$.* As an example, we consider the hypercube setting $\mathbb{X} = \{0, 1\}^D$ similar to Chen and Ying (2024). This setting is natural in practice, as the general discrete space $\mathbb{X} = [S]^D$ which is commonly assumed in many works such as Campbell et al. (2022); Lou et al. (2024); Zhang et al. (2025) can be encoded as a hypercube structure $\{0, 1\}^{D \log |S|}$. We let the eigenvalues of $L = -Q$ be ordered as $0 = \lambda_1 < \lambda_2 \leq \cdots \leq \lambda_M$.

**Assumption 7.** *Let the discrete state space be $\mathbb{X} = \{0, 1\}^D$. For any pair of distinct states $x \neq y$, assume the rate matrix $Q(x, y)$ satisfies:*

$$
Q(x, y) = \begin{cases} 1 & (d(x, y) = 1), \\ 0 & (otherwise), \end{cases}
$$

*where $d(x, y)$ denotes the Hamming distance between $x$ and $y$.*

Under Assumption 7, since the diagonal term satisfies $-Q(x, x) = D$, we obtain $\overline{D} = \mathcal{O}(D)$. Moreover, the following spectral property holds.

**Lemma 4.** *Under Assumption 7, for every $w \in \mathbb{X}$, $h_w(x) := \cos(\pi w^\top x)/\sqrt{M}$ is an eigenvector corresponding to the eigenvalue $2|w|$, where $|w|$ is the number of ones in $w$. In particular, $\lambda_2 = 2$, which is independent of the dimension $D$.*

Based on this lemma, we can derive the convergence of the discrete diffusion model as follows.

**Corollary 1.** *Under the assumptions of Proposition 3 and Assumption 5 to 7, for arbitrary $\delta_0 > 0$, the following error bound holds with probability $1 - 2e^{-t}$:*

$$\mathrm{KL}(p_\delta \| \widehat{q}_{T-\delta}) \lesssim \begin{cases} (n^{-(2(s-1)-\delta_0)/(2s-1)} + t/n) \log^8(M \vee n) & (s \geq 1), \\ \frac{M+t}{n} \log^8(M \vee n) & (s < 1). \end{cases}$$

The proof can be found in Appendix E. We observe that when $s \geq 1$, the convergence rate is essentially $\tilde{\mathcal{O}}(n^{-(2(s-1))/(2s-1)})$, which is independent of the state space size $M$. This rate could be achieved by showing that the eigenvectors in this setting can be represented by cosine functions on the continuous space, and that such trigonometric functions can be efficiently approximated by deep neural networks. On the other hand, when $s < 1$, the convergence rate is $\tilde{\mathcal{O}}(M/n)$, matching the error bound of the naive estimator in Theorem 1. In this case, accurately approximating the score function requires aggregating contributions across the entire state space, which leads to higher estimation complexity.

*Example 2: Discrete graph diffusion process.* Finally, we consider a diffusion process defined on a graph. In this setting, each point of $\mathbb{X}$ is randomly generated on a $d$-dimensional, smooth, closed and connected Riemannian manifold $\mathcal{M} \subset \mathbb{R}^D$ isometrically embedded in $\mathbb{R}^d$ through $\iota : \mathcal{M} \to \mathbb{R}^D$, where each point $x$ obeys the uniform distribution $p$ on $\mathcal{M}$ independently. On this point cloud $\mathbb{X}$, we can define the transition rate matrix as the ordinary *graph Laplacian*: First, let the affinity matrix be $W(x, y) := \frac{k_\sigma(x,y)}{p_\sigma(x)p_\sigma(y)}$ where $k_\sigma(x, y) = \exp\left(-\frac{\|x-y\|^2}{2\sigma^2}\right)$ and $p_\sigma(x) = \sum_{y \in \mathbb{X}} k_\sigma(x, y)$, second, using a diagonal matrix $D \in \mathbb{R}^{\mathbb{X} \times \mathbb{X}}$ defined as $D(x, x) = \sum_{y \in \mathbb{X}} W(x, y)$, let the normalized weight matrix as $A = D^{-1}W$, and finally we define the normalized Graph Laplacian as the transition matrix $Q = \frac{1}{\sigma^2}(A - I)$. The stochastic process corresponding to $Q$ is known as a diffusion process on the graph with the normalized weight matrix $A$. In this setting, the graph Laplacian $Q$ can be considered as a discrete approximation of the Laplace–Beltrami operator $-\Delta_\mathcal{M}$ defined on $\mathcal{M}$, and thus the eigenfunctions and eigenvalues of the Laplace–Beltrami operator provide a good approximation of those of $Q$ (i.e., bounding $\varepsilon$ in Assumption 5) (Dunson et al., 2021). Since the eigenfunctions of $\Delta_\mathcal{M}$ are included in the Sobolev space $W_2^\beta(\mathcal{M})$ with arbitrary $\beta$, we can apply our theorem to derive the following bound. The proof is given in Appendix F.

**Corollary 2.** *Suppose that the same assumptions as Theorem 3 and Assumption 5 and 6 hold, and $s > 1$. Then, if $M$ is sufficiently large so that $M = \Omega(n^{\frac{\max\{1+10/d, 5/2+4/d\}(8d+26)}{(d/\beta+2)(s-1)+1}})$ and $\sigma = \left(\frac{\log(M)}{M}\right)^{\frac{1}{4d+13}}$, then there exists an event with high probability on the realization of $\mathbb{X}$ such that, for arbitrary $\beta$ such that $s - 2\beta/d > 1$, the following error bound holds with probability $1 - 2e^{-t}$:*

$$\mathrm{KL}(p_\delta \| \widehat{q}_{T-\delta}) \lesssim \left(n^{-\frac{2(s-1)}{2(s-1)/(2\beta/d)+2s-1}} + \left(\frac{\log(M)}{M}\right)^{\frac{1}{8+4d}} + \frac{t}{n}\right) \log^8(M \vee n).$$

Therefore, a distribution on a point cloud $\mathbb{X}$ on a smooth manifold $\mathcal{M}$ can be well approximated by the discrete diffusion models that utilize the graph diffusion process induced by the graph Laplacian as the forward process.

## 5 Numerical experiment

To complement our theoretical analysis, we conducted a score-matching experiment that exactly instantiates *Example 1* on the hypercube $\{0, 1\}^D$ with $M := 2^D$. For each $D = 6, 8, 10$ we constructed two distributions using the Hadamard basis $H \in \{\pm M^{-1/2}\}^{M \times M}$:

$$\ell(x) := (Hc)_x, \ p(x) = \frac{e^{\ell(x)}}{\sum_{y=0}^{M-1} e^{\ell(y)}}.$$

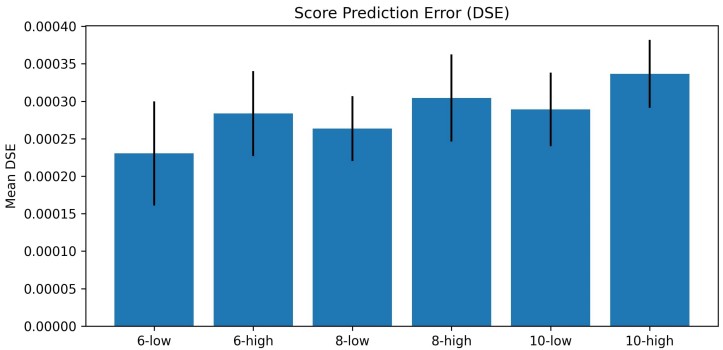

Figure 1: Score matching results on hypercube toy examples. Each bar shows the mean denoising score entropy (DSE) over 10 trials; error bars are $\pm 1$ standard deviation ($\approx 0.4$–$0.7 \times 10^{-4}$).

The coefficient vector $c \in \mathbb{R}^M$ was chosen in two regimes:
**Low-frequency:** $c_0, \ldots, c_4 \sim \mathcal{N}(0, 1)$, $c_k = 0$ for $k \geq 5$.
**High-frequency:** $c_k \sim \mathcal{N}(0, 1)$ for all $k \geq 2$ (DC and first harmonic set to zero).
Activating only the first few modes yields a smooth distribution, whereas using the whole spectrum (minus the DC term) creates a highly oscillatory one. Each distribution was evolved for $t = 1$ under the rate matrix $Q$ shown in Assumption 7. We then train a one-hidden-layer score network $s_\theta(x, y)$ (input $2D$, 256 ReLU units, Softplus output) using the denoising score entropy (DSE) loss. Training used 5000 random Hamming-1 pairs, ADAM (lr $= 10^{-3}$), and 20 epochs. Performance was evaluated as the mean DSE over all Hamming-1 pairs. Each setting was repeated 10 times.

The mild increase from $D = 6$ to $D = 10$ matches the logarithmic dependence on $M = 2^D$ predicted by Theorem 4 and corollary 1, while the higher errors for high-frequency mixtures reflect the bounds' sensitivity to the smoothness parameter. These observations provide concrete evidence that the theoretical guarantees translate directly to practice without any hyper-parameter tuning or architectural changes.

## 6 Conclusion

In this study, we established the first theoretical framework for estimating score functions in discrete diffusion models. We proved that directly approximating the score function for each discrete state using a neural network yields an estimation error rate of $\widetilde{\mathcal{O}}(M/n)$ under a naive analysis and further improved this bound to $\widetilde{\mathcal{O}}(n^{-2\tilde{\beta}(s-1)/(s-1+2s\tilde{\beta}-\tilde{\beta})})$ independent of the state space size $M$ by assuming a polynomial decay condition on the spectral decomposition of the target distribution. Our analysis made use of a decomposition of the target distribution by eigenvectors of the transition matrix $Q$. Then, we utilized the fact that the eigenfunctions can be well approximated by smooth functions (i.e., Besov spaces) on an embedded continuous space in order to reduce the model complexity. We also demonstrated concrete bounds for the hypercube settings and the graph diffusion processes.

One of the main drawbacks of our analysis is that we assumed a mild condition on the score function $s_t^\circ$ such that it is bounded above by $R$ and below by $1/R$. This condition inherits the same condition assumed in the continuous space (Oko et al., 2023). Relaxing this condition to the case where there is no uniform lower bound on the density as performed in Zhang et al. (2024) is an important direction for future work.

### Acknowledgment

SW was partially supported by JST CREST (JPMJCR2115). TS was partially supported by JSPS KAKENHI (24K02905) and JST CREST (JPMJCR2015). This research is supported by the National Research Foundation, Singapore, Infocomm Media Development Authority under its Trust Tech Funding Initiative, and the Ministry of Digital Development and Information under the AI Visiting Professorship Programme (award number AIVP-2024-004). Any opinions, findings and conclusions or recommendations expressed in this material are those of the author(s) and do not reflect the views

of National Research Foundation, Singapore, Infocomm Media Development Authority, and the Ministry of Digital Development and Information.

The authors are grateful to Satoshi Hayakawa for insightful comments, particularly for drawing our attention to the potential ill-definedness of the $\tau$-leaping scheme in the SDE form.

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

# A Construction of Neural Networks

This section collects fundamental tools on function approximation using neural networks. These results play a central role in our approximation error analysis. Many of the lemmas below are based on Oko et al. (2023).

## A.1 Composition and Combination Lemmas

We begin with lemmas that describe how to combine multiple neural networks. These lemmas are essential to realize a large neural network that approximates complicated functions.

**Lemma 5** (Nakada and Imaizumi (2020)). *For any neural networks $\phi^1, \ldots, \phi^k$ with $\phi^i : \mathbb{R}^{d_i} \to \mathbb{R}^{d_{i+1}}$ and $\phi^i \in \Phi(L^i, W^i, S^i, B^i)$, there exists a network $\phi \in \Phi(L, W, S, B)$ such that $\phi(x) = \phi^k \circ \phi^{k-1} \circ \cdots \circ \phi^1(x)$ for $x \in \mathbb{R}^{d_1}$ with*

$$L = \sum_{i=1}^{k} L^i, \quad \|W\|_\infty \leq 2 \sum_{i=1}^{k} \|W^i\|_\infty, \quad S \leq 2 \sum_{i=1}^{k} S^i, \quad B \leq \max_{1 \leq i \leq k} B^i.$$

**Lemma 6** (Oko et al. (2023)). *For any networks $\phi^1, \ldots, \phi^k$ with $\phi^i : \mathbb{R}^{d_i} \to \mathbb{R}^{d'_i}$ and $\phi^i \in \Phi(L^i, W^i, S^i, B^i)$, there exists a network $\phi \in \Phi(L, W, S, B)$ such that $\phi(x) = [\phi^1(x_1)^\top, \ldots, \phi^k(x_k)^\top]^\top : \mathbb{R}^{d_1 + \cdots + d_k} \to \mathbb{R}^{d'_1 + \cdots + d'_k}$ for $x = (x_1^\top \cdots x_k^\top)^\top$ with*

$$L = \max_{1 \leq i \leq k} L^i, \quad \|W\|_\infty \leq 2 \sum_{i=1}^{k} \|W^i\|_\infty,$$

$$S \leq 2 \sum_{i=1}^{k} (S^i + LW_L^i), \quad B \leq \max\{\max_{1 \leq i \leq k} B^i, 1\}.$$

**Lemma 7** (Oko et al. (2023)). *For any networks $\phi^1, \ldots, \phi^k$ with $\phi^i : \mathbb{R}^{d_i} \to \mathbb{R}^d$ and $\phi^i \in \Phi(L^i, W^i, S^i, B^i)$, there exists a network $\phi \in \Phi(L, W, S, B)$ such that $\phi(x) = \sum_{i=1}^{k} \phi^i(x_i) : \mathbb{R}^{d_1 + \cdots + d_k} \to \mathbb{R}^d$ for $x = (x_1^\top \cdots x_k^\top)^\top$ with*

$$L = \max_{1 \leq i \leq k} L^i + 1, \quad \|W\|_\infty \leq 4 \sum_{i=1}^{k} \|W^i\|_\infty,$$

$$S \leq 4 \sum_{i=1}^{k} (S^i + LW_L^i) + 2W_L, \quad B \leq \max\{\max_{1 \leq i \leq k} B^i, 1\}.$$

**Lemma 8** (Oko et al. (2023)). *For $\underline{t}_1 < \underline{t}_2 < \bar{t}_1 < \bar{t}_2$ and $f(x, t) : \mathbb{R}^d \times \mathbb{R} \to \mathbb{R}$, assume that $\phi^i(x, t)$ approximates $f(x, t)$ up to $\varepsilon > 0$ within $[\underline{t}_i, \bar{t}_i]$ $(i = 1, 2)$. There exist networks $\phi_{\text{swit}}^1(t; \underline{t}_2, \bar{t}_1), \phi_{\text{swit}}^2(t; \underline{t}_2, \bar{t}_1) \in \Phi(3, (1, 2, 1, 1)^\top, 8, \max\{\bar{t}_1, (\bar{t}_1 - \underline{t}_2)^{-1}\})$ with*

$$|\phi_{\text{swit}}^1(t; \underline{t}_2, \bar{t}_1)\phi^1(x, t) + \phi_{\text{swit}}^2(t; \underline{t}_2, \bar{t}_1)\phi^2(x, t) - f(x, t)| \leq \varepsilon \quad (t \in [\underline{t}_1, \bar{t}_2]).$$

## A.2 Approximations of rational functions

The following lemmas describe how to approximate elementary operations such as multiplication and reciprocal using neural networks. These are particularly important because the score function in discrete diffusion models involves the ratio of probabilities between two discrete states, which can be approximated by neural networks designed to emulate rational functions.

**Lemma 9** (Oko et al. (2023)). *For $k \geq 2, C \geq 1, 0 < \varepsilon_{\text{error}} \leq 1$ and $\varepsilon > 0$, there exists a network $\phi_{\text{mult}}(x_1, \ldots, x_k) \in \Phi(L, W, S, B)$ with $L = \mathcal{O}(\log k(\log \varepsilon^{-1} + k \log C)), \|W\| = 48k, S = \mathcal{O}(k \log \varepsilon^{-1} + k \log C), B = C^k$ such that $|\phi_{\text{mult}}(x)| \leq C^k, \phi_{\text{mult}}(x_1, \ldots, x_k) = 0$ if at least one of $x_i$ is equal to zero, and*

$$\left| \phi_{\text{mult}}(x_1', x_2', \ldots, x_k') - \prod_{i=1}^{k} x_i \right| \leq \varepsilon + kC^{k-1}\varepsilon_{\text{error}}, \quad \forall x \in [-C, C]^k, \|x - x'\|_\infty \leq \varepsilon_{\text{error}}.$$

**Lemma 10** (Oko et al. (2023)). *For a constant $R > 1$ and any $0 < \varepsilon < R^{-1}$, there exists a network $\phi_{\text{rec}}(x_1, \ldots, x_k) \in \Phi(L, W, S, B)$ with $L = \mathcal{O}(\log^2 \varepsilon^{-1}), \|W\| = \mathcal{O}(\log^3 \varepsilon^{-1}), S = \mathcal{O}(\log^4 \varepsilon^{-1}), B = \mathcal{O}(\varepsilon^{-2})$ such that*

$$\left| \phi_{\text{rec}}(x') - \frac{1}{x} \right| \leq \varepsilon + R^2 |x' - x|, \quad \forall x \in [R^{-1}, R], x' \in \mathbb{R}.$$

## B    Poisson Random Measures with evolving intensity

In this section, we introduce the formal definition of Poisson random measures with evolving intensity, which is a foundational concept required for implementing the $\tau$-leaping algorithm. Although this concept is not central to the main theoretical results of this paper, we include the definition here for completeness and reference.

**Definition 2** (Protter (1983); Ren et al. (2025)). *Consider a probability space $(\Omega, \mathcal{F}, \mathbb{P})$ and a measure space $(X, \mathcal{B}, \nu)$. A non-negative predictable process $\lambda_t(y)$ on $\mathbb{R}_+ \times X \times \Omega$ is assumed to satisfy for any $T > 0$:*

$$\int_0^T \int_{\mathbb{X}} \left( 1 \vee |y| \vee |y|^2 \right) \lambda_t(y) \nu(dy) dt < \infty \quad a.s.$$

*where $\nu$ is the counting measure. probability measure $N[\lambda](\mathrm{d}t, \mathrm{d}y)$ on $\mathbb{R}_+ \times \mathbb{X}$ is called a Poisson random measure with evolving intensity $\lambda_t(y)$ if:*

1. *For any $B \in \mathcal{B}$ and $0 \leq s < t$, $N[\lambda]((s, t] \times B) \sim \mathcal{P}\left( \int_s^t \int_B \lambda_\tau(y) \nu(dy) d\tau \right)$. where $\mathcal{P}(\cdot)$ represents a Poisson distribution with the given expectation.*

2. *For any $t \geq 0$, $\mathcal{B}$ and disjoint sets $\{B_i\}_{i=1}^n$, the processes $\{N_t[\lambda](B_i) := N[\lambda]((0, t] \times B_i)\}_{i=1}^n$ are independent.*

## C    Proof of Theorem 1

To prove Theorem 1, we decompose the total estimation error into approximation and generalization components . We begin by analyzing the score approximation error with a neural network. This part captures the model bias, and the resulting bound depends on the smoothness of the score function and the expressive power of the network class. The subsequent subsection will then address the generalization error using tools from statistical learning theory. Together, these analyses yield the desired estimation error bound.

### C.1    Approximation error analysis

In this subsection, we construct a neural network that approximates the true score function $s_t^\circ(x, y)$ in a compositional manner by separately approximating $p_t(x)$ and its reciprocal $1/p_t(x)$, and then combining them in a multiplicative way.

**Lemma 11.** *For any $\varepsilon_1 > 0$, there exists a neural network $\phi^1(x, t) \in \Phi(L, W, S, B)$ such that*

$$\left| \phi^1(x, t) - M p_t(x) \right| \leq \varepsilon_1 \quad \forall t > 0.$$

*The parameters of $\phi^1(x, t)$ are bounded as follows:*

$$L = \mathcal{O}(\log^2 M \varepsilon_1^{-1}),$$
$$\|W\|_\infty = \mathcal{O}(M \log^3 M \varepsilon_1^{-1}),$$
$$S = \mathcal{O}(M \log^5 M \varepsilon_1^{-1}),$$
$$B = \mathcal{O}(M^2 \vee \log M \varepsilon_1^{-1}).$$

*Proof.* We begin by approximating the function $e^{-\lambda_j t}$ using a neural network. Define $A := \log 3\varepsilon^{-1}$.

For $j = 0$, we set $\phi_j = 1$ so that $e^{-\lambda_j t}$ is approximated without an error. For $j > 0$, using the Taylor expansion, we obtain

$$e^{-\lambda_j t} = e^{-\lambda_j s}\left(\sum_{i=0}^{k}\frac{(-\lambda_j)^i}{i!}(t-s)^i + \frac{(-\lambda_j)^{k+1}}{(k+1)!}(\theta(t-s))^{k+1}\right),$$

where $\theta \in [0, 1]$. Setting $k = \max\{\lceil 2e^2\overline{D}\rceil, \lceil\log 3\varepsilon^{-1}\rceil\}$, we obtain the bound

$$\left|e^{-\lambda_j s}\frac{(-\lambda_j)^{k+1}}{(k+1)!}(\theta(t-s))^{k+1}\right| \leq \left|\frac{(-\lambda_j)^{k+1}}{(k+1)!}\right| < \left|\frac{1}{((k+1)/2e\overline{D})^{k+1}}\right| \leq |e^{-k+1}| \leq \varepsilon.$$

Here, we used $(n/e)^n < n!$ in the second inequality. By Lemma 9, for each $\frac{(-\lambda_j)^i}{i!}(t-s)^i$, there exists a neural network $\phi(t; i)$ that approximates it within an error of $\varepsilon/3(k+1)$ over the interval $s \leq t \leq s + 2/\lambda_j$. The network parameters satisfy:

$$L = \mathcal{O}(\log^2\varepsilon^{-1}),$$
$$\|W\|_\infty = \mathcal{O}(\log\varepsilon^{-1}),$$
$$S = \mathcal{O}(\log^2\varepsilon^{-1}),$$
$$B = \mathcal{O}(1).$$

Similarly, by lemma 7, there exists a neural network $\phi_s(t)$ that approximates $e^{-\lambda_j t}$ within an error of $\varepsilon/3$ over $s \leq t \leq s + 2/\lambda_j$, with the following parameter bounds:

$$L = \mathcal{O}(\log^2\varepsilon^{-1}),$$
$$\|W\|_\infty = \mathcal{O}(\log^2\varepsilon^{-1}),$$
$$S = \mathcal{O}(\log^4\varepsilon^{-1}),$$
$$B = \mathcal{O}(1).$$

Note that $|c_j u_j(x)| \leq 1$. Indeed,

$$c_j = \sum_{x\in\mathbb{X}} p_0(x)u_j(x) \leq \sqrt{\left(\sum_{x\in\mathbb{X}} p_0(x)^2\right)\left(\sum_{x\in\mathbb{X}} u_j(x)^2\right)} = \sqrt{\sum_{x\in\mathbb{X}} p_0(x)^2} \leq \sum_{x\in\mathbb{X}} p_0(x) = 1.$$

Thus, we can construct a neural network $\phi_j^*(x, t)$ that approximates $c_j u_j(x)e^{-t\lambda_j}$ as follows:

$$\phi_j^*(x, t) := \phi_{\mathrm{mult}}(\phi_{\mathrm{swit}}^2(t; 1/\lambda_j, 2/\lambda_j), c_j u_j(x)\phi_0(t)) +$$
$$\sum_{s=1}^{\lceil A\rceil - 1}\phi_{\mathrm{mult}}(\phi_{\mathrm{swit}}^1(t; (s+1)/\lambda_j, (s+2)/\lambda_j), \phi_{\mathrm{swit}}^2(t; s/\lambda_j, (s+1)/\lambda_j), c_j u_j(x)\phi_{s/\lambda_j}(t)).$$

Using Lemma 9, the approximation error from each $\phi_{\mathrm{mult}}$ is bounded by $\varepsilon/\log\varepsilon^{-1}$. We set the parameters of $\phi_{\mathrm{mult}}$ to $L = \mathcal{O}(\log\varepsilon^{-1}), \|W\|_\infty = \mathcal{O}(1), S = \mathcal{O}(\log\varepsilon^{-1}), B = \mathcal{O}(1)$. To restrict the input $t$ of $\phi_j^*(x, t)$ to the range $[0, A]$, we define $\phi_j(x, t) := \mathrm{ReLU}(\phi_j^*(x, t) - \phi_j^*(x, A)) + \phi_j^*(x, A)$. This ensures that $\phi_j(x, t)$ approximates $e^{-\lambda_j t}$ with an error at most $\varepsilon$ for all $t \geq 0$. For $t \leq A$, the following inequality holds: $|\phi_j(x, t) - c_j u_j(x)e^{-\lambda_j t}| \leq \varepsilon/3 + \varepsilon/3 < \varepsilon$. For $t > A$, we have $|\phi_j(x, t) - c_j u_j(x)e^{-\lambda_j t}| \leq |\phi_j(x, t) - \phi_j(x, A)| + |\phi_j(x, A) - c_j u_j(x)e^{-\lambda_j A}| + |c_j u_j(x)(e^{-\lambda_j t} - e^{-\lambda_j A})| \leq 0 + 2\varepsilon/3 + \varepsilon/3 = \varepsilon$. Thus, the parameters of $\phi(x, t)$ are evaluated as follows:

$$L = \mathcal{O}(\log^2\varepsilon^{-1}),$$
$$\|W\|_\infty = \mathcal{O}(\log^3\varepsilon^{-1}),$$
$$S = \mathcal{O}(\log^5\varepsilon^{-1}),$$
$$B = \mathcal{O}(\log\varepsilon^{-1}).$$

Finally, setting $\varepsilon := \varepsilon_1/M^2$ and summing $\phi_j(x,t)$ over all $j = 1,\ldots,M$, we construct a neural network $\phi^1(x,t)$ that approximates $Mp_t(x)$ within an error of $\varepsilon_1$. By Lemmas 7 and 9, the parameter bounds are given by

$$L = \mathcal{O}(\log^2 M\varepsilon_1^{-1}),$$
$$\|W\|_\infty = \mathcal{O}(M \log^3 M\varepsilon_1^{-1}),$$
$$S = \mathcal{O}(M \log^5 M\varepsilon_1^{-1}),$$
$$B = \mathcal{O}(M^2 \vee \log M\varepsilon_1^{-1}).$$

$\square$

Building on this result, we next construct a neural network that approximates the reciprocal $1/p_t(x)$.

**Lemma 12.** *For any $0 < \varepsilon_0 \le \frac{1}{R}$ and $\varepsilon_1 > 0$ there exists a neural network $\phi^2(x,t) \in \Phi(L,W,S,B)$ satisfying the following inequality:*

$$\left| \phi^2(x,t) - \frac{1}{Mp_t(x)} \right| \le \varepsilon_0 + R^2\varepsilon_1 \quad \forall t > 0.$$

*with the parameters of $\phi^2$ are bounded as follows:*

$$L = \mathcal{O}(\log^2 \varepsilon_0^{-1} \vee \log^2 M\varepsilon_1^{-1}),$$
$$\|W\|_\infty = \mathcal{O}(\log^3 \varepsilon_0^{-1} \vee M \log^3 M\varepsilon_1^{-1}),$$
$$S = \mathcal{O}(\log^4 \varepsilon_0^{-1} \vee M \log^5 M\varepsilon_1^{-1}),$$
$$B = \mathcal{O}(M^2 \vee \varepsilon_0^{-2} \vee \log M\varepsilon_1^{-1}).$$

*Proof.* From Lemmas 5 and 10, there exists $\phi^2(x,t) = \phi_{\mathrm{rec}} \circ \phi^1(x,t)$ satisfying

$$\left| \phi^2(x,t) - \frac{1}{Mp_t(x)} \right| \le \varepsilon_0 + R^2\varepsilon_1.$$

Here, the network parameters coincide with those stated above. Regarding $p_t$, the boundedness of the score function ensures that $Mp_t(x) = \frac{Mp_t(x)}{\sum_{y\in\mathbb{X}} p_t(y)} = \frac{M}{\sum_{y\in\mathbb{X}} s_t(y,x)} \ge \frac{1}{R}$ which satisfies the conditions of Lemma 10 $\square$

Combining the approximations of $p_t(x)$ and its reciprocal, we are now ready to construct a neural network that directly approximates the score function $s_t^\circ(x,y) = p_t(y)/p_t(x)$. The following lemma formalizes this construction and provides the approximation error bound.

**Lemma 13.** *For any $0 < \varepsilon_0 \le \frac{1}{R}$ and $\varepsilon_1, \varepsilon_2 > 0$, there exists a network $\phi_{\mathrm{score}}(x,y,t) \in \mathcal{F}$ such that*

$$|\phi_{\mathrm{score}}(x,y,t) - s_t^\circ(x,y)| \lesssim \varepsilon_0 + R^2\varepsilon_1 + \varepsilon_2 \quad \forall t > 0.$$

*Here, the parameters of $\phi_{\mathrm{score}}$ are evaluated as:*

$$L = \mathcal{O}(\log^2 \varepsilon_0^{-1} \vee \log^2 M\varepsilon_1^{-1} \vee \log \varepsilon_2^{-1}),$$
$$\|W\|_\infty = \mathcal{O}(\log^3 \varepsilon_0^{-1} \vee M \log^3 M\varepsilon_1^{-1}),$$
$$S = \mathcal{O}(\log^4 \varepsilon_0^{-1} \vee M \log^5 M\varepsilon_1^{-1} \vee \log \varepsilon_2^{-1}),$$
$$B = \mathcal{O}(\varepsilon_0^{-2} \vee \log M\varepsilon_1^{-1}).$$

*In particular, the following holds:*

$$\int_\delta^T \mathbb{E}_{x\sim p_t} \left[ \sum_{y\neq x} \mathrm{BR}_K(\phi_{\mathrm{score}}(x,y,t)\|s_t^\circ(x,y)) \cdot s_t^\circ(x,y)Q(x,y) \right] dt \lesssim R\log(R)T\overline{D}(\varepsilon_0 + R^2\varepsilon_1 + \varepsilon_2)^2.$$

*Proof.* Consider the neural network $\phi^3(x, y, t) := [\phi^1(y, t), \phi^2(x, t)]^\top$, which parallelizes the estimation of $p_t(y)$ and $1/p_t(x)$. The function $\phi_{\mathrm{score}}(x, y, t) := \phi_{\mathrm{mult}} \circ \phi^3(x, y, t)$ is a neural network that estimates $s_t^\circ(x, y) = \frac{p_t(y)}{p_t(x)}$, and its error is evaluated as follows:

$$|\phi_{\mathrm{score}}(x, y, t) - s_t^\circ(x, y)| \le |\phi_{\mathrm{mult}}(\phi^1(y, t), \phi^2(x, t)) - \phi^1(y, t)\phi^2(x, t)| + |\phi^1(y, t)\phi^2(x, t) - s_t^\circ(x, y)|$$

$$\le \varepsilon_2 + \left|\phi^1(y, t)\phi^2(x, t) - \frac{\phi^1(y, t)}{Mp_t(x)} + \frac{\phi^1(y, t)}{Mp_t(x)} - s_t^\circ(x, y)\right|$$

$$\le \varepsilon_2 + |\phi^1(y, t)|\left|\phi^2(x, t) - \frac{1}{Mp_t(x)}\right| + \left|\frac{1}{Mp_t(x)}\right||\phi^1(y, t) - p_t(y)|$$

$$\le \varepsilon_0 + (R^2 + R)\varepsilon_1 + \varepsilon_2.$$

From Lemmas 5, 9, 11 and 12, the parameters are bounded as showed in above:

$$L = \mathcal{O}(\log^2 \varepsilon_0^{-1} \vee \log^2 M\varepsilon_1^{-1} \vee \log \varepsilon_2^{-1}),$$

$$\|W\|_\infty = \mathcal{O}(\log^3 \varepsilon_0^{-1} \vee M \log^3 M\varepsilon_1^{-1}),$$

$$S = \mathcal{O}(\log^4 \varepsilon_0^{-1} \vee M \log^5 M\varepsilon_1^{-1} \vee \log \varepsilon_2^{-1}),$$

$$B = \mathcal{O}(\varepsilon_0^{-2} \vee \log M\varepsilon_1^{-1}).$$

Considering $\bar{\phi}_{\mathrm{score}}(x, y, t)$ which restricts the output of $\phi_{\mathrm{score}}(x, y, t)$ to $[1/R, R]$, Assumption 3 ensures that this restriction does not increase the error, and the order of the parameters remains unchanged. Thus, we can conclude $\phi_{\mathrm{score}} \in \mathcal{F}$.

Since $1/R \le \phi_{\mathrm{score}}, s^\circ \le R$ and $0 \le x - \log(x) - 1 \le 4\log(R)(x - 1)^2$ for $x \in [-1/R^2, R^2]$, we have that

$$\mathrm{BR}_K(\phi_{\mathrm{score}}(x, y, t)\|s_t^\circ(x, y)) = \left(\frac{\phi_{\mathrm{score}}(x, y, t)}{s_t^\circ(x, y)} - \log \frac{\phi_{\mathrm{score}}(x, y, t)}{s_t^\circ(x, y)} - 1\right)$$

$$\le 4\log(R)(\phi_{\mathrm{score}}(x, y, t)/s_t^\circ(x, y) - 1)^2.$$

Therefore, we obtain that

$$\int_\delta^T \mathbb{E}_{x\sim p_t}\left[\sum_{y\ne x} \mathrm{BR}_K(\phi_{\mathrm{score}}(x, y, t)\|s_t^\circ(x, y)) \cdot s_t^\circ(x, y)Q(x, y)\right]\mathrm{d}t$$

$$\le \int_\delta^T \mathbb{E}_{x\sim p_t}\left[\sum_{y\ne x}(\phi_{\mathrm{score}}(x, y, t)/s_t^\circ(x, y) - 1)^2 s_t^\circ(x, y)Q(x, y)\right]\mathrm{d}t$$

$$\le \int_\delta^T \mathbb{E}_{x\sim p_t}\left[\sum_{y\ne x} 4\log(R)(\phi_{\mathrm{score}}(x, y, t) - s_t^\circ(x, y))^2 s_t^\circ(x, y)^{-1}Q(x, y)\right]\mathrm{d}t$$

$$\le 4R\log(R)(\varepsilon_0 + (R^2 + MR)\varepsilon_1 + \varepsilon_2)^2 \int_\delta^T \overline{D}\mathrm{d}t$$

$$\lesssim R\log(R)T\overline{D}(\varepsilon_0^2 + R^4\varepsilon_1^2 + \varepsilon_2^2),$$

which achieves the assertion. $\qquad\square$

## C.2 Generalization Error Analysis

This subsection focuses on bounding the generalization error. The analysis proceeds in three steps:

- First, we relate the score estimation error to a conditional form via a denoising representation.
- Second, we define a loss class and control its generalization error using local Rademacher complexity and Peeling device.
- Third, we derive explicit complexity bounds using covering number estimates for neural networks.

We begin by expressing the explicit score matching entropy as the denoising score entropy. This representation justifies interpreting the score entropy as the empirical loss.

**Lemma 14** (Lou et al. (2024)). *For any $s_t(x_t, y)$ and $t > 0$, the following holds:*

$$\sum_{x_t} \sum_{y \neq x_t} \mathrm{BR}_K(s_t(x_t, y) \| s_t^\circ(x_t, y)) \cdot s_t^\circ(x_t, y) p_t(x_t) Q(x_t, y)$$

$$= \sum_{x_0} \sum_{x_t} \sum_{y \neq x_t} \mathrm{BR}_K(s_t(x_t, y) \| s_t^\circ(x_t, y \mid x_0)) \cdot s_t^\circ(x_t, y \mid x_0) p_t(x_t \mid x_0) p_0(x_0) Q(x_t, y) + C,$$

*where $C$ is a constant independent of $s_t$.*

*Proof.*

$$\sum_{x_t} \sum_{y \neq x_t} \mathrm{BR}_K(s_t(x_t, y) \| s_t^\circ(x_t, y)) \cdot s_t^\circ(x_t, y) p_t(x_t) Q(x_t, y)$$

$$= - \sum_{x_t, y \neq x_t} \left( \log \frac{s_t(x_t, y)}{s_t^\circ(x_t, y)} - \frac{s_t(x_t, y)}{s_t^\circ(x_t, y)} + 1 \right) \cdot p_t(y) Q(x_t, y)$$

$$= - \sum_{x_0, x_t, y \neq x_t} \left( \log \frac{s_t(x_t, y)}{s_t^\circ(x_t, y \mid x_0)} - \frac{s_t(x_t, y)}{s_t^\circ(x_t, y \mid x_0)} + 1 \right) \cdot p_t(y \mid x_0) p_0(x_0) Q(x_t, y)$$

$$+ \underbrace{\left( \sum_{x_t, y \neq x_t} (\log s_t^\circ(x_t, y)) \cdot p_t(y) Q(x_t, y) - \sum_{x_0, x_t, y \neq x_t} \log s_t^\circ(x_t, y \mid x_0) \right)}_{=:C}$$

$$= \sum_{x_0, x_t, y \neq x_t} \mathrm{BR}_K(s_t(x_t, y) \| s_t^\circ(x_t, y \mid x_0)) \cdot s_t^\circ(x_t, y \mid x_0) \cdot p_t(x_t \mid x_0) p_0(x_0) Q(x_t, y) + C,$$

where $C$ is a constant depending only on $p_t, Q$. $\qquad \square$

Next, We relate the Hellinger distance to the expected excess risk.

**Lemma 15.** *For any $g \in G := \{g := \ell_{\widehat{s}} - \ell_{s^\circ} \mid \widehat{s} \in \mathcal{F}\}$, we have that $\int_0^{T-\delta} h^2(\widehat{s}_t, s_t^\circ) \mathrm{d}t \lesssim Pg$.*
*Proof.* From Lemma 14, we have:

$$Pg = L(\widehat{s}_t) - L(s_t)$$

$$= \int_\delta^T \mathbb{E}_{x_t \sim p_t} \left[ \sum_{y \neq x_t} \mathrm{BR}_K(\widehat{s}_t(x_t, y) \| s_t^\circ(x_t, y)) \cdot s_t^\circ(x_t, y) Q(x_t, y) \right] \mathrm{d}t$$

$$= - \int_\delta^T \mathbb{E}_{x_t \sim p_t} \left[ \sum_{y \neq x_t} \left( \log \frac{\widehat{s}_t(x_t, y)}{s_t^\circ(x_t, y)} - \frac{\widehat{s}_t(x_t, y)}{s_t^\circ(x_t, y)} + 1 \right) \cdot s_t^\circ(x_t, y) Q(x_t, y) \right] \mathrm{d}t$$

$$\geq - \int_\delta^T \mathbb{E}_{x_t \sim p_t} \left[ \sum_{y \neq x_t} \left( 2 \left( \sqrt{\frac{\widehat{s}_t(x_t, y)}{s_t^\circ(x_t, y)}} - 1 \right) - \frac{\widehat{s}_t(x_t, y)}{s_t^\circ(x_t, y)} + 1 \right) \cdot s_t^\circ(x_t, y) Q(x_t, y) \right] \mathrm{d}t$$

$$= \int_\delta^T \mathbb{E}_{x \sim p_t} \left[ \sum_{y \neq x_t} (-2 \sqrt{\widehat{s}_t(x_t, y) s_t^\circ(x_t, y)} + s_t^\circ(x_t, y) + \widehat{s}_t(x_t, y)) Q(x_t, y) \right] \mathrm{d}t$$

$$= \int_\delta^T h(\widehat{s}_t, s_t^\circ)^2 \mathrm{d}t,$$

which concludes the assertion. $\qquad \square$

Here, we aim to bound the generalization error using the local Rademacher complexity $R_n(G_r)$. We define $G_r := \{g := \ell_s - \ell_{s^\circ} \mid g \in G, Pg \leq r\}$.

**Lemma 16.** *For any $g \in G := \{g := \ell_s - \ell_{s^\circ} \mid s \in \mathcal{F}\}$, it holds that $\|g\|_\infty \lesssim TR\overline{D}$ which also indicates that $Pg^2 \lesssim TR\overline{D}Pg$.*

*Proof.* Define $\ell_s^*(x, y, x_0, t) := \mathrm{BR}_K(s_t(x,y)\|s_t^\circ(x, y \mid x_0)) \cdot s_t^\circ(x, y \mid x_0)$. Then, for any $s \in \mathcal{F}$, $x, y \in \mathbb{X}$, it holds that

$$
\begin{aligned}
&|\ell_s^*(x, y, x_0, t) - \ell_{s^\circ}^*(x, y, x_0, t)| \\
&= \left| \left( \frac{s_t(x,y) - s^\circ(x,y)}{s_t^\circ(x, y \mid x_0)} - \log \frac{s_t(x,y)}{s_t^\circ(x,y)} \right) \cdot s_t^\circ(x, y \mid x_0) \right| \\
&= \left| (s_t(x,y) - s^\circ(x,y)) - \log \frac{s_t(x,y)}{s_t^\circ(x,y)} s_t^\circ(x, y \mid x_0) \right| \\
&\leq |s_t(x,y) - s^\circ(x,y)| + \left| \log \frac{s_t(x,y)}{s_t^\circ(x,y)} s_t^\circ(x, y \mid x_0) \right| \\
&\leq 2R + 2\log(R)s_t^\circ(x, y \mid x_0).
\end{aligned}
$$

Therefore, we arrive at

$$
\begin{aligned}
|\ell_s(x_0) - \ell_{s^\circ}(x_0)| &\leq \int_\delta^T \mathbb{E}_{x_t \sim p_t(\cdot|x_0)} \left[ \sum_{y \neq x_t} |\ell_{s^1}^*(x, y, x_0, t) - \ell_{s^2}^*(x, y, x_0, t)|Q(x_t, y) \right] \mathrm{d}t \\
&\leq \int_\delta^T \mathbb{E}_{x_t \sim p_t(\cdot|x_0)} \left[ \sum_{y \neq x_t} [2R + 2\log(R)s_t^\circ(x, y \mid x_0)]Q(x_t, y) \right] \mathrm{d}t \\
&\leq 2(T - \delta)R\overline{D} + \int_\delta^T \mathbb{E}_{y_t \sim p_t(\cdot|x_0)} \left[ \sum_{x \neq y_t} 2\log(R)Q(x, y_t) \right] \mathrm{d}t \\
&\leq 4(T - \delta)R\overline{D},
\end{aligned}
$$

where we used the symmetricity of $Q$ and $R \geq 1$ in the last equality. Therefore, we have $Pg^2 \leq \|g\|_\infty Pg \lesssim TR\overline{D}Pg$. $\qquad\square$

We introduce two classical tools for controlling the supremum of the empirical loss.

**Proposition 17** (Peeling device (van de Geer, 2000; Bartlett et al., 2005; Koltchinskii, 2006))**.**
*Suppose there exists a function $\phi : [0, \infty) \to [0, \infty)$ and $\widehat{r}^* > 0$ such that $\forall r > \widehat{r}^*$,*

$$
\phi(4r) \leq 2\phi(r), \; R_n(G_r) \leq \phi(r).
$$

*Then, $\forall r > \widehat{r}^*$, the following holds:*

$$
\mathbb{E}_{\{\sigma_i\}, \{x_i\}} \left[ \sup_{g \in G} \frac{\frac{1}{n}\sum_{i=1}^n \sigma_i g(x_i)}{Pg + r} \right] \leq \frac{4\phi(r)}{r}.
$$

**Proposition 18** (Talagrand's concentration inequality)**.** *(Talagrand, 1996; Bousquet, 2002)] Let $\widetilde{G}$ be a separable set of measurable functions on the probability space $(X, \mathcal{A}, P)$, and suppose that $\forall g \in \widetilde{G}$,*

$$
\mathbb{E}[g] = 0, \; \mathbb{E}[g^2] = v, \; \|g\|_\infty \leq C.
$$

*Then, $\forall t > 0$,*

$$
\Pr\left\{ \sup_{g \in \widetilde{G}} \frac{1}{n}\sum_{i=1}^n g(x_i) \geq 2\mathbb{E}_{\{x_i'\}}\left[ \sup_{g \in \widetilde{G}} \frac{1}{n}\sum_{i=1}^n g(x_i') \right] + \sqrt{\frac{2tv}{n}} + \frac{2tC}{n} \right\} \leq e^{-t}.
$$

By using these theorems, the prediction error can be decomposed into the approximation error (model bias term) and the generalization error (variance term).

**Lemma 19.** *Define* $s^* := \arg\min_{f \in \mathcal{F}} L(f)$, $\widehat{s} := \arg\min_{f \in \mathcal{F}} \widehat{L}(f)$, $\widehat{g} := \ell_{\widehat{s}} - \ell_{s^\circ}$, *and* $r^* := L(s^*) - L(s^\circ)$. *For the function* $\phi(r)$ *defined in Proposition 17, there exists* $\widehat{r} \gtrsim \max\left\{\phi(\widehat{r}), r^*, \frac{tT\overline{D}}{n}\right\}$ *such that the following holds with probability* $1 - 2e^{-t}$:

$$P\widehat{g} \lesssim \widehat{r} + r^* + \mathcal{O}\left(\frac{TR\overline{D}t}{n}\right).$$

*Proof.* By Lemma 16,

$$Pg^2 \lesssim TR\overline{D}Pg, \quad \|g\|_\infty \le TR\overline{D}.$$

Define $\widetilde{G} := \left\{\widetilde{g} := \frac{Pg - g(x)}{Pg + r} \mid g \in G\right\}$. Then, the following holds:

$$\mathbb{E}[\widetilde{g}] = \frac{Pg - Pg}{Pg + r} = 0,$$

$$\|\widetilde{g}\|_\infty = \left\|\frac{Pg - g}{Pg + r}\right\|_\infty \le \left\|\frac{g}{r}\right\|_\infty + 1 \lesssim \frac{TR\overline{D}}{r},$$

$$\|\widetilde{g}\|_{L_2}^2 = \frac{\|Pg - g\|_{L_2}^2}{(Pg + r)^2} = \frac{Pg^2 - (Pg)^2}{(Pg + r)^2} \le \frac{Pg^2}{2rPg} \lesssim \frac{TR\overline{D}Pg^2}{rPg^2} = \frac{TR\overline{D}}{r}.$$

Using Proposition 17 to bound the term $\mathbb{E}_{\{x_i'\}}\left[\sup_{g \in \widetilde{G}} \frac{1}{n}\sum_{i=1}^n g(x_i')\right]$ in Proposition 18, we have

$$\mathbb{E}\left[\sup_{\widetilde{g} \in \widetilde{G}} |(P - P_n)\widetilde{g}|\right] \le 2R_n(\widetilde{G})$$

$$= 2\mathbb{E}_{\{\sigma_i\},\{x_i\}}\left[\sup_{g \in G} \frac{1}{n}\sum_{i=1}^n \sigma_i \frac{Pg - g(x_i)}{Pg + r}\right]$$

$$= 2\mathbb{E}_{\{\sigma_i\},\{x_i\}}\left[\sup_{g \in G} \frac{1}{n}\sum_{i=1}^n \sigma_i \frac{g(x_i)}{Pg + r}\right]$$

$$\le \frac{8\phi(r)}{r}.$$

The first inequality follows from a standard property of the Rademacher complexity, and the final bound is derived using Proposition 17. Therefore, from Proposition 18, it follows that with probability at least $1 - e^{-t}$,

$$(P - P_n)\left(\frac{\widehat{g}}{P\widehat{g} + r}\right) \le \sup_{\widetilde{g} \in \widetilde{G}} \frac{1}{n}\sum_{i=1}^n \widetilde{g}(x_i) \lesssim \frac{16\phi(r)}{r} + \sqrt{\frac{2tTR\overline{D}}{nr}} + \frac{2tTR\overline{D}}{nr} =: \psi_n(r).$$

Let us define $g^* := l \circ s_t^* - l \circ s_t$. Then, we get

$$P\widehat{g} = L(\widehat{s}_t) - L(s_t)$$

$$= L(\widehat{s}_t) - \widehat{L}(\widehat{s}_t) + \widehat{L}(\widehat{s}_t) - \widehat{L}(s_t^*) + \widehat{L}(s_t^*) - L(s_t^*) + L(s_t^*) - L(s_t) + \widehat{L}(s_t) - \widehat{L}(s_t)$$

$$= (P - P_n)\widehat{g} + (P_n - P)g^* + Pg^*$$

$$\le (P\widehat{g} + r)\psi_n(r) + (P_n - P)g^* + r^*.$$

Since $Pg^{*2} \le TR\overline{D}Pg^*$, the second term can be bounded via Bernstein's inequality (Wainwright, 2019, Proposition 2.14) as

$$(P_n - P)g_* \le \mathcal{O}\left(\sqrt{\frac{r^*t}{n}} + \frac{TR\overline{D}t}{n}\right) \le \mathcal{O}\left(\frac{TR\overline{D}t}{n}\right) + \frac{r^*}{TR\overline{D}},$$

with probability $1 - e^{-t}$ for $t > 0$. We now choose $\widehat{r} \gtrsim \max\left\{\phi(\widehat{r}), r^*, \frac{tTR\overline{D}}{n}\right\}$. If necessary, we can scale $\widehat{r}$ so that $\psi_n(\widehat{r}) \le 1/2$. Under this condition, the following inequality holds:

$$P\widehat{g} \le \frac{\psi_n(\widehat{r})}{1 - \psi_n(\widehat{r})}\widehat{r} + \frac{1 + 1/(TR\overline{D})}{1 - \psi_n(\widehat{r})}r^* + \mathcal{O}\left(\frac{TR\overline{D}t}{n}\right)$$

$$\lesssim \widehat{r} + r^* + \mathcal{O}\left(\frac{TR\overline{D}t}{n}\right).$$

$\square$

To evaluate the complexity function $\phi(r)$, we use known bounds on the covering number of ReLU neural networks.

**Lemma 20** (Schmidt-Hieber (2020)). *For $\mathcal{S} \subset \Phi(L, W, S, B)$, the covering number $\log N(\varepsilon, \mathcal{S}, \|\cdot\|_\infty)$ satisfies*

$$\log N(\varepsilon, \mathcal{S}, \|\cdot\|_\infty) \leq 2SL \log(L(B \vee 1)W\varepsilon^{-1}).$$

Combining Lemma 20 with the Dudley integral gives the following bound on the local Rademacher complexity.

**Lemma 21.** *For $\phi(r)$ defined in Proposition 17,*

$$\phi(r) \leq \mathcal{O}\left(\sqrt{\frac{SLr}{n}} \log(L(B \vee 1)\|W\|_\infty n)\right).$$

By combining these local Rademacher complexity controls, we obtain the proof of Theorem 1.

*Proof of Theorem 1.* From Lemmas 15 and 19, with probability at least $1 - 2e^{-t}$ the following inequality holds:

$$\int_\delta^T h^2(\widehat{s}_t, s_t^\circ)\mathrm{d}t \lesssim \mathcal{O}\left(\widehat{r} + r^* + \frac{TR\overline{D}t}{n}\right).$$

According to Lemma 13, the model bias term $r^*$ can be bounded as

$$r^* \lesssim TR\overline{D}(\varepsilon_0 + \varepsilon_0^{-2}\varepsilon_1 + \varepsilon_2),$$

Furthermore, from Lemma 21, the generalization error $\widehat{r}$ is bounded as

$$\widehat{r} = \mathcal{O}\left(\frac{SL \log(L(B \vee 1)\|W\|_\infty n)}{n}\right).$$

The network parameters satisfy the following upper bounds: $L = \mathcal{O}(\log^2 \varepsilon_0^{-1} \vee \log^2 M\varepsilon_1^{-1} \vee \log \varepsilon_2^{-1})$, $\|W\|_\infty = \mathcal{O}(\log^3 \varepsilon_0^{-1} \vee M \log^3 M\varepsilon_1^{-1})$, $S = \mathcal{O}(\log^4 \varepsilon_0^{-1} \vee M \log^5 M\varepsilon_1^{-1} \vee \log \varepsilon_2^{-1})$, $B = \mathcal{O}(M^2 \vee \varepsilon_0^{-2} \vee \log M\varepsilon_1^{-1})$. To balance the size of each term, we set $\varepsilon_0 = \frac{M}{nD} \wedge \frac{1}{R}, \varepsilon_1 = \frac{M^3}{n^3\overline{D}^3} \vee \frac{M}{n\overline{D}}$, $\varepsilon_2 = \frac{M}{n\overline{D}}$. Given that $\overline{D} = \mathcal{O}(M)$, the desired bound follows. $\square$

# D   Proof of Theorem 3

In this section, we provide the proof of Theorem 3. While the subsequent analysis closely parallels the arguments in Appendix C, the key distinction here lies in the use of function approximation theory in anisotropic Besov spaces.

## D.1   Approximation thoery in anisotropic Besov spaces

Here, we give the function approximation method in an anisotropic Besov class by deep neural networks. We define the affine composition model, which composes affine transformations with functions in an anisotropic Besov space:

$$\mathcal{H}_{\text{aff}} := \{h(Ax + b) \mid h \in U(B_{p,q}^\beta([0,1]^d)),\ A \in \mathbb{R}^{d \times D},\ b \in \mathbb{R}^d \text{ s.t. } Ax + b \in [0,1]^d \ (\forall x \in \Omega)\}.$$

Under this model, the following bound on the approximation error is known.

**Proposition 22** (Suzuki and Nitanda (2021)). *Suppose $x$ follows the uniform distribution on $\Omega = [0,1]^D$, and define $\tilde{x} = Ax + b \in \mathbb{R}$, which is assumed to have a bounded density supported on $[0,1]^d$. Suppose there exists a constant $C$ such that $\|A\|_\infty \vee \|b\|_\infty \leq C$. Then for $0 < p, q, r \leq \infty$ and $\beta \in \mathbb{R}_{++}^d$ satisfying $\widetilde{\beta} > (1/p - 1/r)_+$, the following approximation error bound holds:*

$$R_r(\Phi(L_1(d), W_1(d), S_1(d), d(C+1)B_1(d)), \mathcal{H}_{\text{aff}}) \lesssim N^{-\widetilde{\beta}},$$

*where $R_r(\mathcal{F}, \mathcal{H}) := \sup_{f^* \in \mathcal{H}} \inf_{f \in \mathcal{F}} \|f^* - f\|_{L^r(\Omega)}$ denotes the worst-case approximation error. Here, the network parameters are set as*

$$L_1(d) := 3 + 2 \left\lceil \log_2 \left( \frac{3^{d \vee m}}{\epsilon c_{(d,m)}} \right) + 5 \right\rceil \lceil \log_2(d \vee m) \rceil,$$

$$W_1(d) := NW_0,$$

$$S_1(d) := [(L_1(d) - 1)W_0^2 + 1]N,$$

$$B_1(d) := \mathcal{O}(N^{d(1+\nu^{-1})(1/p-\widetilde{\beta})_+}),$$

*where $\epsilon = N^{-\widetilde{\beta}} \log(N)^{-1}$, and $c_{(d,m)}$ is a constant depending only on $d$ and $m$.*

### D.2 Approximation error bound

Under Assumptions Assumption 5 to 7, we construct a neural network that approximates the score function $s_t^\circ(x, y)$.

**Lemma 23.** *For every $k \in [1, \ldots, M]$, there exists a neural network $\phi_{\mathrm{cont}}^1(x, t) \in \Phi(L, W, S, B)$ such that*

$$\left| \phi_{\mathrm{cont}}^1(x, t) - M p_t(x) \right| \lesssim (N^{-\widetilde{\beta}} \vee e^{-t\lambda_2} \varepsilon) f_1(k) \vee e^{-t\lambda_2} f_2(k) \quad \forall t > 0,$$

*The network parameters satisfy:*

$$L = \mathcal{O}(\log^2(M \vee N)),$$

$$\|W\|_\infty = \mathcal{O}(\log^3 M \vee kN),$$

$$S = \mathcal{O}(\log^5 M \vee kN \log N),$$

$$B = \mathcal{O}(k^\gamma \mathrm{poly}(M \vee N)),$$

*where the functions $f_1(k)$ and $f_2(k)$ are defined as:*

$$f_1(k) \lesssim \begin{cases} k^{1-(s-\gamma)} & (s - \gamma < 1), \\ \log k & (s - \gamma = 1), \\ 1 & (s - \gamma > 1), \end{cases} \quad f_2(k) \lesssim \begin{cases} M^{1-s} & (s < 1), \\ \log M & (s = 1), \\ k^{1-s} & (s > 1), \end{cases}$$

*for $1 \le k \le M$ and $f_2(M) = 0$.*

*Proof.* We aim to approximate the function $M p_t(x) = M \sum_{j=1}^M c_j u_j(x) e^{-t\lambda_j}$ using a neural network by truncating the sum to a fixed number of terms. Specifically, we consider $p_t^*(x) = \sum_{j=1}^k c_j u_j^*(x) e^{-t\lambda_j}$, where each $u_j^* \in \mathcal{H}_{\mathrm{aff}}$. By Proposition 22, there exists a neural network $\phi_j^0 \in \Phi(L_1(d), W_1(d), S_1(d), \gamma_j B_1(d))$ such that

$$\|u_j^* - \phi_j^0\|_{L^\infty(\Omega)} \lesssim \gamma_j N^{-\widetilde{\beta}}.$$

Following the same strategy as in Lemma 11, we can construct a neural network to approximate the exponential decay $e^{-t\lambda_j}$. We define $\phi_j^*(t)$ as follows:

$$\phi_j^*(t) := \phi_{\mathrm{mult}}(\phi_{\mathrm{swit}}^2(t; 1/\lambda_j, 2/\lambda_j), \phi_0(t)) +$$
$$\sum_{s=1}^{\lceil A \rceil - 1} \phi_{\mathrm{mult}}(\phi_{\mathrm{swit}}^1(t; (s+1)/\lambda_j, (s+2)/\lambda_j), \phi_{\mathrm{swit}}^2(t; s/\lambda_j, (s+1)/\lambda_j), \phi_{s/\lambda_j}(t)).$$

The product $u_j^*(x) \cdot e^{-t\lambda_j}$ can be approximated using $\phi_j^*(t)$ and $\phi_j^0$, and by summing over $j = 1, \ldots, k$ and multiplying by $M$, we can construct the neural network $\phi_{\mathrm{cont}}^1 \in \Phi(L, W, S, B)$ satisfying $\|\phi_{\mathrm{cont}}^1 - M p_t^*\| \le N^{-\widetilde{\beta}} \sum_{j=1}^k \gamma_j j^{-s}$. Here, we set the approximation error of $\phi_j^*(t)$ and $\phi_{\mathrm{mult}}$ to $\varepsilon = M^{-2} N^{-\widetilde{\beta}}$. According to Lemmas 7 and 9, the parameters of $\phi_{\mathrm{cont}}^1$ can be bounded as follows:

$$L = \mathcal{O}(\log^2(M \vee N)),$$

$$\|W\|_\infty = \mathcal{O}(\log^3 M \vee kN),$$

$$S = \mathcal{O}(\log^5 M \vee kN \log N),$$

$$B = \mathcal{O}(\gamma_k \mathrm{poly}(M \vee N)).$$

Now, the total error in approximating $p_t(x)$ is given by

$$|Mp_t(x) - \phi_{\text{cont}}^1(x,t)| \leq N^{-\widetilde{\beta}} \sum_{j=1}^{k} \gamma_j j^{-s} + M \left| \sum_{j=1}^{M} c_j u_j(x) e^{-t\lambda_j} - \sum_{j=1}^{k} c_j u_j^*(x) e^{-t\lambda_j} \right|$$

$$\lesssim N^{-\widetilde{\beta}} \sum_{j=1}^{k} j^{-(s-\gamma)} + \sqrt{M}\varepsilon \sum_{j=1}^{k} c_1 j^{-s} e^{-t\lambda_j} + M \left| \sum_{j=k+1}^{M} c_1 j^{-s} u_j(x) e^{-t\lambda_j} \right|.$$

For the sum $\sum_{j=1}^{k} j^{-(s-\gamma)}$ and $\sum_{j=k+1}^{M} j^{-s}$, we have

$$\sum_{j=1}^{k} j^{-(s-\gamma)} \leq 1 + \int_1^k j^{-(s-\gamma)} \mathrm{d}j \lesssim \begin{cases} k^{1-(s-\gamma)} & (s-\gamma < 1), \\ \log k & (s-\gamma = 1), \\ 1 & (s-\gamma > 1), \end{cases}$$

$$\sum_{j=k+1}^{M} j^{-s} \leq \int_k^M j^{-s} \mathrm{d}j \lesssim \begin{cases} M^{1-s} & (s < 1), \\ \log M & (s = 1), \\ k^{1-s} & (s > 1). \end{cases}$$

Since $c_1 = 1/\sqrt{M}$, we can achieve the desired upper bound. $\qquad \square$

**Lemma 24.** *For any $0 < \varepsilon_0 \leq R^{-1}$, there exists a neural network $\phi_{\text{cont}}^2(x,t) \in \Phi(L, W, S, B)$ such that*

$$\left| \phi_{\text{cont}}^2(x,t) - \frac{1}{Mp_t(x)} \right| \lesssim \varepsilon_0 + R^2((N^{-\widetilde{\beta}} \vee e^{-t\lambda_2}\varepsilon)f_1(k) \vee e^{-t\lambda_2} f_2(k)) \quad \forall t > 0,$$

*with network parameters bounded as:*

$$L = \mathcal{O}(\log^2 \varepsilon_0^{-1} \vee \log^2(M \vee N)),$$
$$\|W\|_\infty = \mathcal{O}(\log^3(M \vee \varepsilon_0^{-1}) \vee kN),$$
$$S = \mathcal{O}(\log^5 M \vee \log^4 \varepsilon_0^{-1} \vee kN \log N),$$
$$B = \mathcal{O}(\varepsilon_0^{-2} \vee \gamma_k \text{poly}(M \vee N)).$$

*Here, $f_1(k)$ and $f_2(k)$ are defined in Lemma 23.*

*Proof.* As in Lemma 12, we can construct $\phi_{\text{cont}}^2(x,t) = \phi_{\text{rec}} \circ \phi_{\text{cont}}^1(x,t)$ to achieve the desired approximation. $\qquad \square$

**Lemma 25.** *Let $0 < \varepsilon_0 \lesssim R^{-1}$ and $N \in \mathbb{N}$. Then, there exists a neural network $\phi_{\text{cont}}(x,y,t) \in \mathcal{F}$ such that*

$$|\phi_{\text{cont}}(x,y,t) - s_t^\circ(x,y)| \lesssim \varepsilon_0 + R^2((N^{-\widetilde{\beta}} \vee e^{-t\lambda_2}\varepsilon)f_1(k) \vee e^{-t\lambda_2} f_2(k)) \quad \forall t > 0.$$

*The parameters of $\phi_{\text{cont}}$ are bounded as follows:*

$$L = \mathcal{O}(\log^2 \varepsilon_0^{-1} \vee \log^2(M \vee N)),$$
$$\|W\|_\infty = \mathcal{O}(\log^3(M \vee \varepsilon_0^{-1}) \vee kN),$$
$$S = \mathcal{O}(\log^5 M \vee \log^4 \varepsilon_0^{-1} \vee kN \log N),$$
$$B = \mathcal{O}(\varepsilon_0^{-2} \vee \gamma_k \text{poly}(M \vee N)).$$

*Here, $f_1(k)$ and $f_2(k)$ are defined in Lemma 23. Moreover, the following bound holds:*

$$\int_\delta^T \mathbb{E}_{x \sim p_t} \left[ \sum_{y \neq x} \text{BR}_K(\phi_{\text{cont}}(x,y,t) \| s_t^\circ(x,y)) \cdot s_t^\circ(x,y) Q(x,y) \right] \mathrm{d}t$$

$$\lesssim R \log(R) T \overline{D} (\varepsilon_0 + R^2 \lambda_2^{-1}(\varepsilon f_1(k) \vee f_2(k)) + R^2 N^{-\widetilde{\beta}} f_1(k))^2.$$

*Proof.* The proof proceeds in a similar manner to Lemma 13. We construct a network $\phi_{\text{cont}}^3(x, y, t) := [\phi_{\text{cont}}^1(y, t), \phi_{\text{cont}}^2(x, t)]^\top$ by combining the neural networks that approximate $Mp_t(y)$ and $1/Mp_t(x)$ in parallel. Then, define the overall network as $\phi_{\text{cont}}(x, y, t) := \phi_{\text{mult}} \circ \phi^3(x, y, t)$. This network estimates the true score $s_t^\circ(x, y) = \frac{p_t(y)}{p_t(x)}$, and the error can be bounded as follows:

$$|\phi_{\text{cont}}(x, y, t) - s_t^\circ(x, y)| \le |\phi_{\text{mult}}(\phi^1(y, t), \phi^2(x, t)) - \phi^1(y, t)\phi^2(x, t)| + |\phi^1(y, t)\phi^2(x, t) - s_t^\circ(x, y)|$$

$$\le \varepsilon_1 + \left| \phi^1(y, t)\phi^2(x, t) - \frac{\phi^1(y, t)}{Mp_t(x)} + \frac{\phi^1(y, t)}{Mp_t(x)} - s_t^\circ(x, y) \right|$$

$$\le \varepsilon_1 + |\phi^1(y, t)| \left| \phi^2(x, t) - \frac{1}{Mp_t(x)} \right| + \frac{1}{Mp_t(x)} |\phi^1(y, t) - Mp_t(y)|$$

$$\lesssim \varepsilon_0 + \varepsilon_1 + R^2((N^{-\widetilde{\beta}} \vee e^{-t\lambda_2}\varepsilon)f_1(k) \vee e^{-t\lambda_2} f_2(k)).$$

Setting $\varepsilon_1 = \varepsilon_0$ gives the desired approximation rate.
By the same argument as Lemma 13, we can bound the divergence as

$$\int_\delta^T \mathbb{E}_{x \sim p_t} \left[ \sum_{y \ne x} \text{BR}_K(\phi_{\text{cont}}(x, y, t) \| s_t^\circ(x, y)) \cdot s_t^\circ(x, y) Q(x, y) \right] dt$$

$$\le \int_\delta^T \mathbb{E}_{x \sim p_t} \left[ \sum_{y \ne x} 4R \log(R) |\phi_{\text{cont}}(x, y, t) - s_t^\circ(x, y)|^2 Q(x, y) \right] dt$$

$$\le 4R \log(R) \int_\delta^T (\varepsilon_0 + R^2((N^{-\widetilde{\beta}} \vee e^{-t\lambda_2}\varepsilon)f_1(k) \vee e^{-t\lambda_2} f_2(k)))^2 \overline{D} dt$$

$$\lesssim R \log(R) \left( T\overline{D}\varepsilon_0^2 + R^4 \int_\delta^{t_0} (e^{-t\lambda_2}(\varepsilon f_1(k) \vee f_2(k)))^2 \overline{D} dt + R^4 \int_{t_0}^T (N^{-\widetilde{\beta}} f_1(k))^2 \overline{D} dt \right)$$

$$\lesssim R \log(R) T\overline{D}(\varepsilon_0 + R^2 \lambda_2^{-1}(\varepsilon f_1(k) \vee f_2(k)) + R^2 N^{-\widetilde{\beta}} f_1(k))^2,$$

where we define $t_0 := \min \left\{ \delta, \lambda_2^{-1}(\widetilde{\beta} \log N + \log(\varepsilon \vee \frac{f_2(k)}{f_1(k)})) \right\}$. □

### D.3 Proof of the statement

By combining the results from Appendices C.2 and D.2, we are now ready to prove Theorem 3.

*Proof of Theorem 3.* By applying Lemmas 15 and 19, we obtain

$$\int_\delta^T h^2(\widehat{s}_t, s_t^\circ) dt \lesssim \mathcal{O}\left( \widehat{r} + r^* + \frac{TR\overline{D}t}{n} \right),$$

with probability $1 - 2e^{-t}$. By Lemmas 21 and 25, the model bias term $r^*$ and the generalization error $\widehat{r}$ can be bounded as:

$$r^* \lesssim TR \log(R) \overline{D}[\varepsilon_0 + R^2 \lambda_2^{-1}(\varepsilon f_1(k) \vee f_2(k)) + R^2 N^{-\widetilde{\beta}} f_1(k)]^2,$$

$$\widehat{r} = \mathcal{O}\left( \frac{SL \log(L(B \vee 1) \|W\|_\infty n)}{n} \right).$$

By setting $N = \mathcal{O}(nk^{-1}f_1(k)^2)^{\frac{1}{1+2\widetilde{\beta}}}$ and $\varepsilon_0 = \mathcal{O}(n^{-1/2})$, we have

$$r^* + \widehat{r} \lesssim \left[ (n^{-\widetilde{\beta}} k^{\widetilde{\beta}} f_1(k))^{\frac{2}{1+2\widetilde{\beta}}} + \lambda_2^{-2} f_2(k)^2 + \lambda_2^{-2} \varepsilon^2 f_1(k)^2 \right] \log^8(M \vee N).$$

We consider balancing the first and second terms as follows:

1. When $s \le 1$ (and thus $s - \gamma \le 1$): By setting $k = M$, we obtain

$$r^* + \widehat{r} \lesssim \left( n^{-\frac{2\widetilde{\beta}}{1+2\widetilde{\beta}}} M^{\frac{2(1-(s-\gamma)+\widetilde{\beta})}{1+2\widetilde{\beta}}} + \lambda_2^{-2} \varepsilon^2 M^2 \right) \log^8(M \vee N).$$

2. When $1 < s$ and $s - \gamma \leq 1$: By setting $k = (\lambda_2^{1+2\widetilde{\beta}} n^{-\widetilde{\beta}})^{-\frac{1}{(1+2\widetilde{\beta})(s-1)+(1-(s-\gamma)+\widetilde{\beta})}}$, we have that

$$r^* + \widehat{r} \lesssim \left( \lambda_2^{-\frac{2(1-(s-\gamma)+\widetilde{\beta})}{(1+2\widetilde{\beta})(s-1)+(1-(s-\gamma)+\widetilde{\beta})}} n^{-\frac{2\widetilde{\beta}(s-1)}{(s-1)(1+2\widetilde{\beta})+(1-(s-\gamma)+\widetilde{\beta})}} \right.$$

$$\left. + \varepsilon^2 \lambda_2^{-\frac{2(s-1+1-(s-\gamma))(1+2\widetilde{\beta})+2(1-(s-\gamma)+\widetilde{\beta})}{(1+2\widetilde{\beta})(s-1)+(1-(s-\gamma)+\widetilde{\beta})}} n^{\frac{2\widetilde{\beta}(1-(s-\gamma))}{(s-1)(1+2\widetilde{\beta})+(1-(s-\gamma)+\widetilde{\beta})}} \right) \log^8(M \vee N).$$

3. When $1 < s$ and $1 < s - \gamma$: By setting $k = (n^{-\widetilde{\beta}} \lambda_2^{1+2\widetilde{\beta}})^{\frac{1}{1+\widetilde{\beta}-s-2s\widetilde{\beta}}}$, we have

$$r^* + \widehat{r} \lesssim \left( \lambda_2^{-\frac{2\widetilde{\beta}}{s-1+2s\widetilde{\beta}-\widetilde{\beta}}} n^{-\frac{2\widetilde{\beta}(s-1)}{s-1+\widetilde{\beta}+2\widetilde{\beta}(s-1)}} + \lambda_2^{-2} \varepsilon^2 \right) \log^8(M \vee N).$$

$\square$

# E Proof of Corollary 1

## E.1 Proof of Lemma 4

Before proceeding to the main proof, we present a preparatory result on the spectral properties of the graph Laplacian in Lemma 4. Since $h_w(x) = (-1)^{w^\top x}$, for every $w \in \{0, 1\}^D$, the function $h_w : \{0, 1\}^D \to \{-1, 1\}$ is an eigenfunction of the adjacency matrix $A$ defined on $\{0, 1\}^D$. Let the state $v_i \in \{0, 1\}^D$ satisfy $d(v_i, x) = 1$ in terms of the Hamming distance, where the index $i$ indicates the coordinate at which $v_i$ and $x$ differ. Then, we obtain

$$\sum_{i=1}^D h_w(v_i) = (D - |w|) h_w(x) - |w| h_w(x) = (D - 2|w|) h_w(x),$$

which implies $h_w$ is an eigenfunction of $A$ with the eigenvalue $D - 2|w|$. Since $L = DI - A$, where $I$ denotes the identity matrix, it follows that $h_w$ is also an eigenfunction of $L$ with eigenvalue $2|w|$, and in particular, $\lambda_2 = 2$.

## E.2 Main proof

First, we show $h_w(x) := \cos(\pi w^\top x)/\sqrt{M}$ belongs to the Sobolev class $H^\beta([0, 1]^D)$ for any constant $\beta \in \mathbb{N}$. Sobolev spaces are defined by

$$H^\beta(\Omega) := \left\{ f \in L^2(\Omega) \,\middle|\, \|f\|_{H^\beta} := \left( \sum_{|\alpha| \leq \beta} \|D^\alpha f\|_2^2 \right)^{1/2} < \infty \right\}, \quad D^\alpha f := \frac{\partial^{|\alpha|}}{\partial^{\alpha_1} \dots \partial^{\alpha_D}} f.$$

Here $\alpha = (\alpha_1, \dots \alpha_D) \in \mathbb{N}_0^D$. Using the chain rule, we obtain

$$D^\alpha h_w(x) = \frac{1}{\sqrt{M}} p_\alpha(\cos(\pi w^\top x), \sin(\pi w^\top x))(\pi w)^{|\alpha|}.$$

The term $p_\alpha$ is a finite linear combination of trigonometric functions and $|p_\alpha| = \mathcal{O}(1)$. In particular, we have the bound $|D^\alpha h_w| \lesssim \mathcal{O}(|w|^{|\alpha|}/\sqrt{M})$. Thus, the Sobolev norm of $h_w$ is bounded as

$$
\begin{aligned}
\|h_w\|_{H^\beta}^2 &= \sum_{|\alpha| \leq \beta} \|D^\alpha h_w\|_2^2 \\
&= \sum_{|\alpha| \leq \beta} \int_{[0,1]^D} |D^\alpha h_w|^2 dx \\
&= \sum_{|\alpha| \leq \beta} \mathcal{O}(|w|^{2|\alpha|}/M) \\
&= \sum_{k=0}^{\beta} \sum_{|\alpha|=k} \mathcal{O}(|w|^{2|\alpha|}/M) \\
&= \mathcal{O}(D^{3\beta}/M) \\
&= \mathcal{O}(1).
\end{aligned}
$$

This implies that $h_w \in H^\beta([0,1]^D)$ and, in particular, $h_w$ lies in the scaled Sobolev unit ball: there exists a constant $\gamma_w > 0$ such that $h_w \in \gamma_w U(H^\beta)$, where $U(H^\beta)$ denotes the unit ball in $H^\beta(\mathbb{R}^D)$. Since $H^\beta = B_{2,2}^\beta$ in the sense of Besov spaces, Theorem 4 can be applied directly. Specifically, for the function $u_j^*(x) = \cos(\pi w_j^\top x)$, Assumptions 5 and 6 hold with parameters $\varepsilon = 0$, $\gamma_j = \mathcal{O}(1)$, and $\gamma = 0$. Moreover, Lemma 4 ensures that $\lambda_2 = \mathcal{O}(1)$. By choosing $\beta$ sufficiently large, the desired bound follows from Theorem 4.

# F    Proof of Corollary 2

Here, let $\phi_j(x)$ be the eigenfunctions of the Laplace-Beltrami operator $-\Delta_\mathcal{M}$ normalized in $L^2(\mathcal{M})$, and their corresponding eigenvalues are denoted by $0 = \mu_1 < \mu_2 \leq \cdots$, that is, $-\Delta_\mathcal{M}\phi_j = \mu_j\phi_j$, and let $\sigma(-\Delta_\mathcal{M}) := \{\mu_i\}_{i=1}^\infty$ be the set of eigenvalues. For a vector $u = (u(x))_{x \in \mathbb{X}}$, we define

$$
\|u\|_{\ell^2(\hat{p})} := \sqrt{\frac{|\mathbb{S}^{d-1}|\sigma^d}{d} \sum_{x \in \mathbb{X}} \frac{u(x)^2}{\mathcal{N}(x)}},
$$

where $\mathcal{N}(x) := \{y \in \mathbb{X} \mid \|x - y\| \leq \sigma\}$. Then, the following result is known:

**Proposition 26** (Dunson et al. (2021)). *Suppose that $M$ is sufficiently large such that*

$$
M > \max\left(\left(\frac{\mathscr{X}_2 + \mu_k^{d/2+5}}{\min(\Gamma_k, 1)}\right)^{8d+26}, (\mathscr{X}_3 + \mu_k^{(5d+7)/4})^{8d+26}\right), \tag{5}
$$

*where $\mathscr{X}_2, \mathscr{X}_3 > 1$ are constants depending on $d$ and the volume, the radius, the curvature and the second fundamental form of the manifold $\mathcal{M}$, and $\Gamma_k = \min_{1 \leq j \leq k} \text{dist}(\mu_j, \sigma(-\Delta_\mathcal{M})\backslash\{\mu_j\})$. Let $\sigma = \left(\frac{\log(M)}{M}\right)^{\frac{1}{4d+13}}$. Then for $1 \leq j \leq K$, it holds that*

$$
\max_{x \in \mathbb{X}} \left|\frac{u_j(x)}{\|u_j\|_{\ell^2(\hat{p})}} - \phi_j(x)\right| \leq C_2 \left(\frac{\log(M)}{M}\right)^{\frac{1}{8d+16}}. \tag{6}
$$

Since $-\Delta_\mathcal{M}\phi_j = \mu_j\phi_j$, the Sobolev norm of $\phi_j$ for $\beta \in \mathbb{N}$ can be evaluated as

$$
\|\phi_j\|_{W^\beta(\mathcal{M})} \leq C(1 + \|\Delta_\mathcal{M}^\beta \phi_j\|_{L^2(\mathcal{M})}) \leq C(1 + \mu_j^\beta).
$$

Moreover, since it is known that $\mu_j = \Theta(j^{2/d})$ (Hassannezhad et al., 2016), we obtain that

$$
\|\phi_j\|_{W^\beta(\mathcal{M})} \lesssim j^{2\beta/d}.
$$

This implies that we may choose $\gamma = 2\beta/d$. Now suppose that $\beta$ is chosen so that $s - \gamma > 1$. More precisely, we can approximate $\phi_j$ by a composite function $\tilde{\phi}_j \circ \psi_\mathcal{M}$ where $\psi_\mathcal{M} : \mathbb{R}^D \to \mathbb{R}^d$

represents a smooth function giving the local coordinate of $\mathcal{M}$ and $\tilde{\phi}_j$ is a function on this local coordinate that corresponds to $\phi_j$. Then, we may think $\tilde{\phi}_j$ is a function in a Besov space $B_{2,2}^{\beta}$ on a compact domain of $\mathbb{R}^d$, and thus $\tilde{\phi}_j \circ \psi_{\mathcal{M}}$ can be approximated by ReLU deep neural networks with an error $\mathcal{O}(j^{\gamma} N^{-\beta/d})$.

Since $p$ is the uniform distribution, a measure concentration inequality (Wainwright, 2019) implies

$$\frac{|\mathbb{S}^{d-1}|\sigma^d}{d} \frac{1}{\mathcal{N}(x)} = \Theta\left(\frac{1}{M}\right),$$

uniformly over $x \in \mathbb{X}$, with high probability, provided that $\sigma \geq \left(\frac{\log(M)}{M}\right)^{\frac{1}{4d+13}}$. In this event, we see that $\|u\|_{\ell^2(\hat{p})} = \Theta(\|u\|/\sqrt{M}) = \Theta(1/\sqrt{M})$. Then, by Eq. (6), Assumption 5 is satisfied for $u_j^*(x) = \|u_j\|_{\ell^2(\hat{p})}\phi_j(x) \simeq \phi_j(x)/\sqrt{M}$ with the approximation error

$$\varepsilon = \left(\frac{\log(M)}{M}\right)^{\frac{1}{8d+16}}.$$

To achieve the assertion in Theorem 4 with $\widetilde{\beta} = \beta/d$, we choose $k$ in its proof as

$$k = (n^{-\beta/d}\lambda_2^{1+2\beta/d})^{\frac{1}{1+\beta/d-s-2s\beta/d}} \simeq n^{\frac{1}{(d/\beta+2)(s-1)+1}},$$

which yields $\mu_k = \mathcal{O}(n^{\frac{2/d}{(d/\beta+2)(s-1)+1}})$. Hence, $(\mu_k^{d/2+5})^{8d+26} = \mathcal{O}(n^{\frac{(1+10/d)(8d+26)}{(d/\beta+2)(s-1)+1}})$ and $(\mu_k^{(5d+7)/4})^{8d+26} = \mathcal{O}(n^{\frac{(5/2+4/d)(8d+26)}{(d/\beta+2)(s-1)+1}})$. We assume that $M$ is sufficiently large to satisfy Eq. (5) with this choice of $k$, which ensures the approximation error bound above holds.

Based on these arguments, Theorem 4, with $\widetilde{\beta} = \beta/d$ and $p = q = 2$, gives that

$$\mathrm{KL}(p_\delta\|\widehat{q}_{T-\delta}) \lesssim \left(n^{-\frac{2(s-1)}{2(s-1)/(2\beta/d)+2s-1}} + \left(\frac{\log(M)}{M}\right)^{\frac{1}{8+4d}} + \frac{t}{n}\right)\log^8(M \vee n),$$

which yields the assertion.

