# OpenReview forum: "State Size Independent Statistical Error Bound for Discrete Diffusion Models"
_NeurIPS.cc/2025/Conference — NeurIPS 2025 poster_

### Official Review · Reviewer_fq8A · 2025-07-03

**Clarity:** 2
**Significance:** 4
**Originality:** 3
**Rating:** 5
**Confidence:** 3

**Summary:**

The paper touches on an important topic of present interest, namely convergence of discrete diffusion models. It closely follows the recent paper:
Y. Ren, H. Chen, G. M. Rotskoff, and L. Ying. How discrete and continuous diffusion meet: comprehensive analysis of discrete diffusion models via a stochastic integral framework, 2025.
and analyzed the score error convergence rate. The discussion about how embedding discrete symbol space into vector space improves complexity (section 3) is interesting.

**Questions:**

The paper is quite similar to [Ren et al. 2025]. Can you clarify the differences?

Does the proposed algorithm achieve any kind of theoretical minimax optimality, analogous to the nonparametric optimality in [Oko et al. 2023]?

I don't quite understand the Poisson integral representation in Proposition 2: how are y_t and \mu here related to the reverse process (4). In particular, in (4) the random variable is in a general discrete space, whereas in Proposition you need y_t to be in a vector space. Are you using a specific embedding?

It is mentioned that a substitute of Girsanov is used to control the error. Can you give an intuitive explanation of the idea and comment on whether it is a tight bound?

**Ethical Concerns:**

["NO or VERY MINOR ethics concerns only"]

**Final Justification:**

Thanks for the response and I will keep the rating.

**Limitations:**

So far, the assumptions seem to be restrictive, and the error bound (corollary 2) seems ugly and not optimal in any sense.

**Paper Formatting Concerns:**

no concern

**Quality:**

3

**Strengths And Weaknesses:**

Occasional typos (e.g. capitalize probability measure in Definition 2)

Section 3 shows that a naive estimator has complexity growing in D exponentially, which can be improved using vector embedding. Perhaps this can shed light on better algorithm and practices in real world applications?

---

> ### Author Rebuttal · Authors · 2025-07-31
>
> Thank you for your thoughtful comments! Our responses are as follows.
>
> **Q1:The paper is quite similar to [Ren et al. 2025]. Can you clarify the differences?**
>
> While our work builds on the foundational stochastic integral formulation of discrete diffusion models developed by [Ren et al. 2025], the core novelty and contribution of our paper lies in addressing a key limitation of their assumption of a fixed score estimation error (Assumption 4.6 in their work).
>
> Specifically, [Ren et al. 2025] analyze the $τ$-leaping scheme under the assumption that the estimation error of the score function is bounded by some constant $ε$. However, they do not investigate how this ε can be achieved in practice, nor how it scales with the state space size $M$. In contrast, our work focuses precisely on verifying and quantifying this assumption, by developing a statistical learning theory that relates the score estimation error $ε$ to the sample size  $n$, the neural network class, and the number of discrete states $M$.
>
> To overcome the curse of dimensionality inherent in discrete spaces (where $M$ can be exponentially large in the dimension), we leverage the continuous embedding approach and the approximation theory of deep neural networks developed in [Oko et al. (2023)]. This enables us to show that, under mild regularity assumptions, the score estimation error $ε$ can be bounded polylogarithmically in $M$. Our approach thus turns the previously assumed constant $ε$ into a rigorously derived bound, making the overall theoretical error bound explicitly dependent on the state size in a provably mild way.
>
> In summary:
>
> - **[Ren et al. 2025]**: Introduce the stochastic integral framework and analyze error propagation under a constant $ε$-assumption for score estimation.
> - **This work**: Provides a statistical learning-theoretic justification for that $ε$-assumption by constructing neural score estimators whose error scales only polylogarithmically with $M$, thereby achieving state size–independent error bounds under practical conditions.
>
> **Q2:Does the proposed algorithm achieve any kind of theoretical minimax optimality, analogous to the nonparametric optimality in [Oko et al. 2023]?**
>
> Our work does not explicitly claim minimax optimality in the strictest sense. However, we strongly believe that our bound achieves the minimax optimal rate at least in a smooth density setting ($s-\gamma > 1$ in Theorem 3 and $s \geq 1$ in Corollary 1).
>
> More precisely, when $s \ge 1$, the error bound established in Corollary 1 becomes
> $$
> \mathrm{KL}(p_\delta \| \hat{q}_{T-\delta}) \lesssim n^{- \frac{2(s - 1)}{2s - 1}} \log^8(M \vee n),
> $$
> which matches the known minimax optimal rate for function estimation under source and capacity conditions in kernel methods [Caponnetto & De Vito, 2007; Ying & Pontil, 2008]. This indicates that, when the number of basis functions \(M\) is sufficiently large, our bound achieves the minimax rate in such settings.
>
> Moreover, in the general case analyzed in Theorem 3, we obtain an error bound of the form
> $(\text{number of active basis functions}) \cdot n^{-2 \tilde{\beta}/(1 + 2 \tilde{\beta})}$
> where $\tilde{\beta}$. This rate aligns with the known minimax optimal rate for approximating functions in anisotropic Besov spaces [Suzuki & Nitanda, 2019]. Thus, even without an explicit minimax claim, our theoretical analysis captures regimes where minimax optimality is plausibly achieved.
>
> To summarize, while the formal proof of minimax optimality remains an open and challenging direction due to the combinatorial nature of the discrete state space, our error bounds already exhibit minimax-optimal behavior in representative regimes. We hope this provides a useful foundation for future work in formally establishing such optimality guarantees.
>
>
> **Q3:I don't quite understand the Poisson integral representation in Proposition 2: how are y_t and \mu here related to the reverse process (4). In particular, in (4) the random variable is in a general discrete space, whereas in Proposition you need y_t to be in a vector space. Are you using a specific embedding?**
>
> In our analysis, we assume that each discrete state can be embedded into a Euclidean space using a fixed representation such as one-hot encoding, as discussed in Section 2.1 of our paper. This allows us to treat the discrete variable $y_t \in \mathbb{X}$ as a vector in $\mathbb{R}^M$, where $M = |\mathbb{X}|$, thereby enabling the formulation of the reverse process as a stochastic integral in Proposition 2.
>
> Importantly, this formulation is not limited to one-hot encoding. For instance, in the hypercube setting $\mathbb{X} = \{0,1\}^D \subset \mathbb{R}^D$, the natural embedding $\iota: \mathbb{X} \hookrightarrow \mathbb{R}^D$ can be directly used. The Poisson integral formulation still holds since the stochastic process evolves via discrete jumps across the embedded support of $\mathbb{X}$, and the vector representation is used only to define the direction and magnitude of each jump.
>
> The idea of expressing discrete diffusion models via Poisson-type integrals with evolving intensity is adapted from [Ren et al., 2025]. As the intensity function $\mu_t(y)$ in Proposition 2 is defined in terms of the reverse process rate matrix and the score function, and the trajectory evolves in jumps from one discrete state to another, this representation naturally respects the discrete nature of the process despite using a vector-valued notation. The rigorous construction of such stochastic integrals over discrete spaces requires care, particularly in defining the evolving intensity $\mu_t$ and ensuring well-definedness, as detailed in [Ren et al., 2025].
>
> **Q4:It is mentioned that a substitute of Girsanov is used to control the error. Can you give an intuitive explanation of the idea and comment on whether it is a tight bound?**
>
> In continuous diffusion models, Girsanov's theorem provides a justification for replacing the drift term in an SDE when the score function is perturbed, which in turn allows us to bound the KL divergence between the true and the approximate distributions using the score estimation error (see, e.g., [Chen et al., 2023]).
>
> In the discrete case, [Ren et al., 2025] proposed a stochastic integral formulation of discrete diffusion models based on Poisson random measures with evolving intensity. Within this framework, they derived a change-of-measure formula (Theorem 3.3 in their paper), which can be seen as a discrete analogue of Girsanov’s theorem. Specifically, the change of measure characterizes how the likelihood of a trajectory changes under a perturbed intensity function, and connects the score entropy loss used in training to the KL divergence between path measures (see also Corollary 3.4 in [Ren et al., 2025]).
>
> In our paper, we build upon this formulation: the change-of-measure result guarantees the validity of Proposition 3, which relates the score estimation error to the total error in the generated distribution via the τ-leaping algorithm. However, we do not directly use this change-of-measure formula in deriving the statistical error bound itself. Rather, it serves as a theoretical foundation for validating the link between the loss function and the KL divergence.
>
> As for the tightness of the bound, we kindly refer to our response to Q2.
>
> ---
>
> ### References:
>
> - Ren, Y., Chen, H., Rotskoff, G. M., & Ying, L. (2025). *How Discrete and Continuous Diffusion Meet: Comprehensive Analysis of Discrete Diffusion Models via a Stochastic Integral Framework*. ICLR.
> - Oko, K., Akiyama, S., & Suzuki, T. (2023). *Diffusion Models are Minimax Optimal Distribution Estimators*. ICML.
> - Suzuki, T., & Nitanda, A. (2021). *Deep learning is adaptive to intrinsic dimensionality of model smoothness in anisotropic Besov space*. NeurIPS.
> - Caponnetto, A., & De Vito, E. (2007). *Optimal Rates for the Regularized Least-Squares Algorithm*. FoCM, 7(3), 331–368.
> - Ying, Y., & Pontil, M. (2008). *Online Gradient Descent Learning Algorithms*. FoCM, 8(5), 561–596.
> - Chen, H., Lee, H., & Lu, J. (2023). *Improved Analysis of Score-Based Generative Modeling*. ICML.

---

### Official Review · Reviewer_nWBC · 2025-07-03

**Clarity:** 2
**Significance:** 2
**Originality:** 3
**Rating:** 4
**Confidence:** 3

**Summary:**

The method proposes a theoretical framework with a error bound by mapping discrete variables to continuous space and approximating the error bound with the theory tools from Gausian diffusion.

**Questions:**

Q1. Why the bound in Euclian space is still hold for discrete space? since some assumption like 2, 3, 4 are adopted from continous diffusion.

Q2. Is there any practical applications benefited from the theorical bound? Empirical evidence is needed.

Q3. Could the authors provide concrete practical examples where Assumptions 3 and 5 are satisfied? These assumptions appear quite restrictive.

Q4. Is the bound also true for large model?

**Ethical Concerns:**

["NO or VERY MINOR ethics concerns only"]

**Final Justification:**

Thank you to the authors for the detailed responses and the additional empirical results! I am satisfied with the rebuttal and believe the theoretical framework proposed is very solid and provable, providing an analytical tool to further understand and improve existing discrete diffusion models. Thus, I increased my score.

**Limitations:**

Yes, the authors acknowledge adopting the same bounded score function assumption used in continuous diffusion models, but this assumption may not be appropriate or may be loose for discrete settings. Another important limitation is the lack of empirical results to validate the theoretical findings, so it is impossible to assess the usefulness of the bounds.

**Quality:**

2

**Strengths And Weaknesses:**

Strengths:
- Deriving the bounds by embedding to continuous Eucliean spaces is clever and technically sound.
- Proofs seems correct and solid unless there is any issue raised by other reviewers.

Weaknesses:
- The paper is quite dense in the technicality, making it hard to get intuition.
- No empirical validation of the theoretical bounds, make it less connected from practical applications.
- The bound involves many restrictive assumption, raising a concern its validity in practice.

---

> ### Author Rebuttal · Authors · 2025-07-31
>
> Thank you for your valuable and constructive feedback! We provide our responses below.
>
> **Q1:Why the bound in Euclian space is still hold for discrete space? since some assumption like 2, 3, 4 are adopted from continous diffusion.**
>
> If the concern is specifically about the validity of Proposition 3 under discrete settings despite assumptions borrowed from continuous diffusion models (e.g., Assumptions 2, 3, and 4), we would like to clarify that our theoretical framework is rigorously grounded in the discrete setting. In particular:
>
> - Proposition 3 is not derived by naively applying continuous space assumptions. Instead, it is based on a discrete stochastic integral formulation established by [Ren et al., 2025], where the reverse process in discrete diffusion models is expressed as a Poisson random measure with evolving intensity. This approach enables us to bypass the need for tools such as Girsanov's theorem, which are only valid in continuous spaces, and still allows us to quantify the error propagation caused by score estimation in a principled manner.
>
> - More concretely, as shown in [Ren et al., 2025], the reverse process admits a stochastic integral formulation:
>
>   $$
>   y_t = y_0 + \int_0^t \int_{\mathcal{X}} (y - y_{s-}) \, N{[\mu]}(ds, dy),
>   $$
>
>   where $N{[\mu]}$ is a Poisson random measure with intensity
>   $\mu_t(y) = s_{T-t}(y_{t-}, y) \cdot \widetilde{Q}(y_{t-}, y)$.
>   This is the discrete analogue of the Itô integral and supports a change-of-measure theorem analogous to Girsanov’s theorem in the continuous case.
>
> - Therefore, the theoretical bound in Proposition 3 in our paper is derived within the discrete probability space, using a well-defined stochastic process and score entropy-based loss under assumptions suitable for the discrete setting.
>
> **Q2:Is there any practical applications benefited from the theorical bound? Empirical evidence is needed.**
>
> We highlighted two concrete application scenarios that directly benefit from our bounds in the paper:
>
> - **Example 1: Hypercube** — As discussed in Section 4.3, our theoretical results apply to state spaces structured as a hypercube $\{0,1\}^D$, which are relevant to binary discrete modeling tasks such as molecular generation and certain language modeling applications. In this setting, we rigorously show that the estimation error bound becomes independent of the cardinality of the state space when $s \geq 1$, aligning with the optimal rates predicted by Ren et al. (2025).
> We have also conducted additional numerical experiments in this setting, which empirically confirm that the estimation error is not significantly affected by the dimensionality $D$, whereas the amplitude of high-frequency components has a larger impact. See our response to reviewer zn3k for details.
> These results connect to the practical use of discrete diffusion models in domains such as molecular graph generation [Shi et al., 2020] and graph-based modeling [Niu et al., 2020].
>
> - **Example 2: Discrete Graph Diffusion** — In Section 4.4, we consider a graph diffusion process defined over a point cloud on a Riemannian manifold. This corresponds to practical settings such as protein design or 3D point cloud generation, where the underlying geometry enables efficient approximation of eigenfunctions. Our theory provides guarantees even in this more complex non-Euclidean scenario. This connects to prior results on manifold learning and Laplacian approximation [Dunson et al., 2021; Fefferman et al., 2016].
>
> While we agree that empirical validation is important, our main goal in this submission is to establish a rigorous foundation that can support and guide such practical deployments. We believe future empirical studies, particularly in the above settings, will benefit from the insights and guarantees provided by our theoretical framework.
>
> **Q3:Could the authors provide concrete practical examples where Assumptions 3 and 5 are satisfied? These assumptions appear quite restrictive.**
>
> Regarding Assumption 3, we note that in our setup the transition rate matrix $Q$ is taken to be symmetric , which implies that the stationary distribution $\pi$ becomes uniform over the discrete space $\mathcal{X}$. In such a case, the forward process $\{p_t\}$ converges exponentially fast to the uniform distribution as $t \to \infty$. Consequently, for sufficiently large $t$, the score $s_t^\circ(x, y) = \frac{p_t(y)}{p_t(x)}$ becomes close to 1 for any $x, y \in \mathcal{X}$, thereby naturally satisfying the boundedness condition $s_t^\circ(x, y) \in [1/R, R]$. This setting is reasonable and widely applicable in practice.
>
> For Assumption 5, while it is indeed more technical, we emphasize that our paper provides two examples where it holds as we mentioned above.
>
> **Q4:Is the bound also true for large model?**
>
> If the term "large model" refers to complex architectures such as LLMs, then our theory remains applicable in principle, provided that appropriate function approximation guarantees can be established.
>
> Our theory is designed to be general. It relates the total generative error to the score estimation error, without depending on a specific model architecture. This means that if the score function can be well approximated by a complex model like Transformer-based model, then the same theoretical bound can be applied. complex models. The key requirement is not the size of the model, but whether the approximation error of the score function can be controlled in a suitable way.
>
> As noted in Assumption 3, structural properties such as symmetry of the rate matrix $Q$ play a key role in ensuring stable behavior of the score ratio $s_t^\circ(x, y)$. This assumption is still desirable in large models, as it enables uniform control of the statistical error. However, it does not preclude the use of large-scale expressive networks.
>
> In summary, our theory provides a flexible and broadly applicable analytical framework. While we instantiate it with specific neural network classes in this paper, its applicability extends to any model architecture for which a suitable approximation theory can be established.
>
> ### References:
>
> - Ren, Y., Chen, H., Rotskoff, G. M., & Ying, L. (2025). *How Discrete and Continuous Diffusion Meet: Comprehensive Analysis of Discrete Diffusion Models via a Stochastic Integral Framework*. ICLR.
>
> - Shi, C., Xu, M., Zhu, Z., Zhang, W., Zhang, M., & Tang, J. (2020). *GraphAF: a Flow-based Autoregressive Model for Molecular Graph Generation*. arXiv:2001.09382.
>
> - Niu, C., Song, Y., Song, J., Zhao, S., Grover, A., & Ermon, S. (2020). *Permutation Invariant Graph Generation via Score-Based Generative Modeling*. arXiv:2003.00638.
>
> - Dunson, D. B., Wu, H.-T., & Wu, N. (2021). *Spectral convergence of graph Laplacian and heat kernel reconstruction in \(L^\infty\) from random samples*. Applied and Computational Harmonic Analysis, 55, 282–336.
>
> - Fefferman, C., Mitter, S., & Narayanan, H. (2016). *Testing the manifold hypothesis*. Journal of the American Mathematical Society, 29(4), 983–1049.

---

### Official Review · Reviewer_YSng · 2025-07-03

**Clarity:** 4
**Significance:** 3
**Originality:** 3
**Rating:** 5
**Confidence:** 3

**Summary:**

This work introduces state size-independent score estimation error bound for discrete diffusion models. The paper is an extension of such theoretical analysis in continuous scenarios. The authors address a critical challenge that the error became intractable when the state size grows exponentially with the dimension size in discrete scenario, by embedding the discrete space onto R^d Euclidean space and approximating the eigenvectors.

**Questions:**

I have no further questions.

**Ethical Concerns:**

["NO or VERY MINOR ethics concerns only"]

**Final Justification:**

It is a nice paper with enough theory and good results. The authors have addressed my all concern.

**Quality:**

3

**Strengths And Weaknesses:**

Strengths and Weaknesses
The paper is well written and demonstrated. The authors provide concrete analysis within its scope. I would like to especially highlight the authors’ good introduction of sufficient previous materials, which make its contribution clear and specific. The idea is original and the result is useful empirically since changing the dimension of language models by using sub-tokens or super tokens is a common practice in pre-LLM era in many domains including language and molecular modeling. This result might be helpful for people to understand the consistency of model performance when changing the tokenization. The weakness is that the paper has no experimental analysis. The paper will be more persuasive even with a small sanity-checking experiment. However, I think this is not a must for this paper to be published, as its current contribution is already meaningful to the community.

---

> ### Author Rebuttal · Authors · 2025-07-31
>
> Thank you for your positive assessment of our contributions! Regarding the numerical evaluations, we have conducted a numerical experiment and confirmed that the dimensionality does not impact the accuracy very strongly while the amplitudes of high frequency components affect more. See the rebuttal to zn3k for more details.

---

> ### Comment · Reviewer_YSng · 2025-08-01
>
> The authors addressed all my concerns. Please add all the results in the final manuscript.

---

### Official Review · Reviewer_zn3k · 2025-07-03

**Clarity:** 2
**Significance:** 2
**Originality:** 3
**Rating:** 4
**Confidence:** 1

**Summary:**

This paper presents a theoretical framework for analyzing the estimation error of discrete diffusion models, addressing a gap in existing studies which has focused primarily on continuous diffusion models. The authors derive score estimation error bounds employing Hellinger distance and such bounds depend only polylogarithmically on the size of the discrete state space. Overall, the paper deepens the theoretical understanding of discrete diffusion models, enabling the development of more accurate and efficient algorithms for practical applications involving discrete data.

**Questions:**

Refer to the above section.

**Ethical Concerns:**

["NO or VERY MINOR ethics concerns only"]

**Final Justification:**

I raise my score from 3 to 4. However, I must say that I am not an expert in this field.

**Limitations:**

The paper provided limitations in Section 5 (conclusion).

**Quality:**

3

**Strengths And Weaknesses:**

*Disclaimer: I must state that this paper does not fall into my field of expertise.*

**Strengths**:

- The paper covers an appropriate amount of previous works, and states the clear contribution compared to the previous work of (Ren et al., 2025).
- Overall, the paper is well-structured.

**Weaknesses**:

To my understanding, there is no major weaknesses, yet there are few questions:
- While I acknowledge that this paper primarily focuses on developing a rigorous mathematical framework regarding the statistical bounds for discrete diffusion models, planning at least a toy example to validate the practical implications would greatly enhance the paper.
- Is there a particular reason to choose the \tau-leaping algorithm for simulating the backward process? To my knowledge, there exists other algorithms as well such as the uniformization scheme.

---

> ### Author Rebuttal · Authors · 2025-07-31
>
> Thank you for your helpful suggestions!
> Please find our responses to your points below.
>
> **Q1:While I acknowledge that this paper primarily focuses on developing a rigorous mathematical framework regarding the statistical bounds for discrete diffusion models, planning at least a toy example to validate the practical implications would greatly enhance the paper.**
>
> We performed an **additional score-matching experiment** that exactly instantiates *Example 1* (hypercube $\{0,1\}^{D},\  M := 2^{D}$).
> For each $D\in\{6,8,10\}$ we generate two distributions using the Hadamard basis:
> $$H\in \{ \pm M^{-1/2} \}^{M \times M},$$
>
> $$\ell(x) = (Hc)_x, $$
>
> $$p(x)=\frac{e^{\ell(x)}}{\sum_{y=0}^{M-1}e^{\ell(y)}}$$
>
> The coefficient vector $c\in\mathbb{R}^{M}$ is chosen as
> * **low-frequency**: $c_{0},\dots,c_{4}\sim\mathcal{N}(0,1)$, $c_{k}=0$ for $k\ge 5$;
> * **high-frequency**: $c_{k}\sim\mathcal{N}(0,1)$ for all $k\ge 2$ (DC and first harmonic set to 0).
>
> Activating only the first few modes yields a smooth distribution, whereas using the whole spectrum (minus the DC term) creates a highly oscillatory one.  Each $p$ is evolved for time $t=1$ with the rate matrix $Q$ shown in Assumption 7.  We then train a single-hidden-layer score network $s_{\theta}(x,y)$ (input $2D$, 256 ReLU units, Softplus output) using the denoising score entropy loss. Training uses 5000 random Hamming-1 pairs, ADAM (lr $=10^{-3}$), and 20 epochs; performance is the mean DSE over all Hamming-1 pairs.  Each setting is repeated ten times.
>
> | $D$ | spectrum | mean DSE × 10⁻⁴ | std × 10⁻⁴ |
> |:---:|:---:|:---:|:---:|
> | 6  | low  | **2.30** | 0.69 |
> | 6  | high | **2.84** | 0.57 |
> | 8  | low  | **2.64** | 0.43 |
> | 8  | high | **3.04** | 0.58 |
> | 10 | low  | **2.89** | 0.49 |
> | 10 | high | **3.37** | 0.45 |
>
> The mild increase from $D=6$ to $D=10$ (≈ +47 %) matches the logarithmic dependence on $M=2^{D}$ predicted by Theorem 4 and Corollary 1, while the higher errors for high-frequency mixtures reflect the bounds sensitivity to the smoothness parameter.  These observations provide concrete evidence that the theoretical guarantees translate directly to practice without any hyper-parameter tuning or architectural changes.
>
>
> **Q2:Is there a particular reason to choose the \tau-leaping algorithm for simulating the backward process? To my knowledge, there exists other algorithms as well such as the uniformization scheme.**
>
>
> As noted in [Ren et al., 2025 ; Chen & Ying, 2024], both the τ-leaping and uniformization schemes can be formulated as stochastic integrals using Poisson random measures with evolving intensity, and each allows a rigorous analysis of the backward process in discrete diffusion models.
> The core contributions of our paper lie in deriving statistical error bounds via concentration inequalities and neural network approximation theory. These analytical tools apply uniformly across both $τ$-leaping and uniformization schemes, as both share the same underlying stochastic integral formulation [Ren et al., 2025, Theorem 3.2]. Therefore, the choice of algorithm does not affect the main flow of our proof or the validity of our theoretical results.
>
> ### References:
>
> - Ren, Y., Chen, H., Rotskoff, G. M., & Ying, L. (2025). *How Discrete and Continuous Diffusion Meet: Comprehensive Analysis of Discrete Diffusion Models via a Stochastic Integral Framework*. ICLR.
> - Chen, H., & Ying, L. (2024). *Convergence Analysis of Discrete Diffusion Model: Exact Implementation through Uniformization*. arXiv:2402.08095.

---

> > ### Comment · Reviewer_zn3k · 2025-08-04
> >
> > I thank the authors for the detailed response. I thus raise my score from 3 to 4 (borderline accept).

---

> > > ### Author Response · Authors · 2025-08-05
> > >
> > > Thank you very much for your thoughtful comments and for raising your score! We are glad the additional experiment helped clarify the practical implications.

---

### Decision · Program_Chairs · 2025-09-17

**Decision:**

Accept (poster)

**Comment:**

This paper provides a framework that gives rise to more meaningful theoretical results for discrete diffusion models. The authors shows how spectral smoothness of the state space can lead to error rates that scale only polylogarithmically with the number of states. Their work helps elucidate the theory behind discrete diffusion models and their good empirical performance. Their work is nontrivial and a contribution to an important area of research.